# State-dependent impact of major volcanic eruptions observed in ice-core records of the last glacial period

Johannes Lohmann[1], Jiamei Lin[1], Bo M. Vinther[1], Sune O. Rasmussen[1], and Anders Svensson[1]

[1]Physics of Ice, Climate and Earth, Niels Bohr Institute, University of Copenhagen, Denmark

**Correspondence:** Johannes Lohmann (johannes.lohmann@nbi.ku.dk)

**Abstract.** Recently, a record of large, mostly unknown volcanic eruptions occurring during the younger half of the last glacial period (12-60 ka) has been compiled from ice-core records. In both Greenland and Antarctica these eruptions led to significant deposition of sulfate aerosols, which were likely transported in the stratosphere, thereby inducing a climate response. Here we report the first attempt to identify the climatic impact of volcanic eruptions in the last glacial period from ice cores. Average
negative anomalies in high-resolution Greenland and Antarctic oxygen isotope records suggest a multi-annual volcanic cooling. Due to internal climate variability, glaciological noise, as well as uncertainties in the eruption age, the high-frequency noise level often exceeds the cooling induced by individual eruptions. Thus, cooling estimates for individual eruptions cannot be determined reliably. The average isotopic anomaly at the time of deposition also remains uncertain, since the signal degrades over time as a result of layer thinning and diffusion, which act to lower the resolution of both the oxygen isotope and sulfur
records.

    Regardless of these quantitative uncertainties, there is a clear relationship of the magnitude of isotopic anomaly and sulfur deposition. Further, the isotopic signal during the cold stadial periods is larger in Greenland and smaller in Antarctica than during the milder interstadial periods for eruptions of equal sulfur deposition magnitude. In contrast, the largest reductions in snow accumulation associated with the eruptions occur during the interstadial periods. This may be the result of a state-
dependent climate sensitivity, but we cannot rule out that changes in the sensitivity of the isotope thermometer or in the radiative forcing of eruptions of a given sulfur ejection may play a role as well.

## 1   Introduction

Several studies on ice-core and tree-ring records, as well as climate models show that volcanism plays a major role in generating the climate variability observed in the Common Era (PAGES 2k Consortium, 2019). During this period, all of the most
pronounced episodes of reduced tree growth in composite tree ring records can be associated with large volcanic eruptions and their tropospheric cooling effect due to the ejection of sulfur aerosols (Sigl et al., 2015). This suggests that volcanic eruptions are responsible for the strongest multi-annual summer temperature decreases in mid- to high-latitude regions of the Northern Hemisphere. On longer time scales, clusters of large eruptions coincide with centennial cold periods during the Holocene similar to the Little Ice age, as shown in tree ring (Helama et al., 2021) and ice-core records (Kobashi et al., 2017). In climate

model simulations of the past millennium, the temperature variability due to volcanic forcing exceeds the variability due to solar forcing (Schurer et al., 2014), as well as the internal multi-decadal variability (Mann et al., 2021).

    Large future eruptions are unpredictable hazardous perturbations, which may compound the increasing climate extremes that stress ecosystems and societies, and which may increase risks of potential tipping points (Lenton et al., 2008). However, the impact of very large eruptions on the climate is not understood in detail, and it may change with the average state of

the climate. In particular, the climatic impact may differ from glacial to interglacial conditions, or the warmer world of the next centuries. Modeling studies investigating eruptions under future warming scenarios, have reported both an enhanced (Fasullo et al., 2017) and a reduced (Hopfcroft et al., 2018) volcanic cooling, or a change in cooling that depends on the eruption magnitude (Aubry et al., 2021). Another modeling study found no evidence for a difference in the global temperature response during last glacial maximum and present-day conditions Ellerhoff et al. (2022). These contrasting results may be

due to different biases in feedbacks, or missing physics that could be responsible for a potential real-world state dependency. Detailed and direct observations are needed to complement modeling results. But even the largest eruptions of the satellite-era are not large compared to eruptions that eventually occur over time spans of a hundred years or more. Thus, the impact of such eruptions needs to be reconstructed by paleoclimate proxy records that go beyond the observational period. It is challenging to obtain such records with sufficient temporal resolution and accurate dating. Ice cores arguably provide the most detailed

records covering time scales of years up to several hundred millennia. This is because the temporal resolution of the material is large compared to other common stratigraphic archives, which often allows for a layer-counted time scale.

    The ejection of sulfate aerosols into the stratosphere by large volcanic eruptions leads to a sharp peak in polar ice-core sulfate records with a delay of roughly 1-2 years (Burke et al., 2019). Based on the integrated sulfate concentration in Greenland and Antarctic ice cores, continuous records of volcanic eruptions along with rough estimates of the magnitude of the eruptions can

be constructed (Zielinski et al., 1997; Castellano et al., 2004; Gao et al., 2007; Sigl et al., 2015, 2022). Here we use two recently compiled datasets: First, a record of volcanic eruptions in the period 12-60 ka with sulfate peaks detected simultaneously in Greenland and Antarctica (Svensson et al., 2020). Second, continuous records of volcanic eruptions detected in either Greenland or Antarctic ice cores (Lin et al., 2022). The former represents significant volcanic eruptions, which most likely distributed sulfate aerosols globally in the stratosphere and are thus expected to have global climatic impact. The latter is a

much larger set that also includes eruptions with more regional aerosol distribution.

    By analyzing eruptions during the time interval 12-60 ka and comparing them to large historic eruptions, we provide a first attempt of using ice-core data to quantify the cooling effect of very large eruptions with return periods of hundreds of years and more. To this end, sulfate-derived records of volcanic eruptions are combined with high-resolution $\delta^{18}O$ records from the same ice cores. $\delta^{18}O$ is a widely used proxy of surface temperature at the accumulation site, which can be measured with

up to sub-annual time resolution. The variability at such short time scales may not represent reliable climatic information, however, because the original temperature signal is altered by post-depositional processes (Münch et al., 2016). These result in high-frequency noise, referred to as stratigraphic or glaciological noise, as well as a smoothing of short-term anomalies. It is unknown how much climatic information remains at sub-decadal time scales in the glacial ice-core record (Vinther et al., 2010). Here we compare the average short-term cooling signal of a large number of volcanic eruptions to the non-volcanic

proxy variability. This provides insights into the signal preservation of the $\delta^{18}O$ proxy that are useful for future studies on ice-core data of increasingly high-resolution. There are large quantitative uncertainties in the calibration of the $\delta^{18}O$ temperature proxy in the glacial period, which make it difficult to estimate the volcanic cooling in absolute terms. Thus, we complement our analysis with direct ice-core observations of changes in (annual) snow/water accumulation following the detected eruptions, which are not subjected to an unknown calibration. Snow accumulation is known as a climate-sensitive parameter on the large ice sheets, and reductions in precipitation are expected after large volcanic eruptions (Robock and Liu, 1994; Bala et al., 2008).

The glacial volcanic record also allows us to assess a potential state dependency of the climate response, since it features the so-called Dansgaard-Oeschger (DO) cycles. These are abrupt regime shifts in between quasi-stable colder and milder Northern Hemisphere climate conditions, known as Greenland stadials (GS) and Greenland interstadials (GI). The glacial climate resides in these quasi-stable states for centuries up to several millennia. Using different subsets of eruptions, we investigate how the volcanic $\delta^{18}O$ anomaly depends on the climate background state, as well as the sulfate deposition magnitude of the eruptions. This yields observational evidence that complements ongoing investigations into the state dependency of the climate sensitivity (Caballero and Huber, 2013; Köhler et al., 2015; von der Heydt et al., 2016; Ashwin and von der Heydt, 2020).

## 2 Methods and Materials

### 2.1 Records of volcanism

We investigate two records of volcanic eruptions. First, we study the 82 volcanic eruptions identified simultaneously in Greenland and Antarctic ice cores by Svensson et al. (2020) in the period 12-60 ka. These are referred to as *bipolar* eruptions hereafter. Due to difficulties in matching Greenland and Antarctic ice cores around the time of the last glacial maximum, this data set has a gap from 16.5 - 24.5 ka. Further, in Svensson et al. (2020) not all eruptions could be identified with a sulfate spike in all ice cores under consideration. It is unknown whether in these cases the eruption did not yield any sulfate deposition at the ice core site, whether the sulfate deposition was wiped away by snow redistribution, or whether the missing eruption is due to limitations in data resolution and the synchronization procedure. It is likely that highly localized phenomena strongly influence the amount of sulfate deposition that is preserved. A previous study showed that even large events such as the 1815 CE Tambora eruption can be entirely missing in several of a handful of very close-by replicate cores (Gautier et al., 2016). Since for our study a precise alignment of the $\delta^{18}O$ records to the sulfate spikes is crucial, only those ice cores enter our analysis where a given eruption has been identified.

The second data set is a record of volcanic sulfate depositions in either Greenland or Antarctic ice cores in the period 9-60 ka compiled by Lin et al. (2022), which we restrict to the glacial period 11.7-60 ka. This data set consists of the depth of several hundred eruptions in the NGRIP ($N = 780$), NEEM ($N = 311$), GISP2 ($N = 282$), EDC ($N = 211$), WAIS ($N = 470$), and EDML ($N = 470$) ice cores, along with estimated magnitudes derived from the integrated sulfate deposition in the respective cores. Due to differences in resolution and quality of the underlying sulfate records (see Sec. 2.2 and Sec. S1), some ice cores allow for the identification of a larger number of eruptions with smaller average sulfate peaks compared to other cores (Fig. S3). Thus, for instance, the NGRIP data set contains many more small eruptions compared to NEEM.

There are also large differences in the estimated sulfate deposition of different cores for the same eruptions. Reasons for this include actual differences in deposition quantity due to the different locations on the ice sheet, local relative differences in wet and dry deposition, differences in the sulfate measurement method and resolution, post-depositional snow redistribution, as well as potential biases in the thinning function used for the different ice cores. Lin et al. (2022) give one composite volcanic record for each Greenland and Antarctica with 1019 and 691 eruptions, respectively. There, a Greenland Ice Core Chronology 2005 (GICC05) age has been derived from the sulfate spikes in one or more ice cores, and a large subset of eruptions has been matched within cores of the same Hemisphere. In this data set, the sulfate deposition estimate is given individually for all cores where the eruption has been identified, and as an average over the cores.

In the subset of eruptions that were matched in two or more cores of one Hemisphere the scatter of deposition values for individual eruptions in different cores is large (Fig. S4), and the mean deposition values can differ significantly. For instance, on average, the same eruptions in NEEM and GISP2 have a larger estimated sulfate deposition compared to NGRIP. As a result, when cores with larger sulfate deposition only occasionally contribute to the calculation of the average deposition values given by Lin et al. (2022), a certain degree of statistical noise is introduced. But since the differences in the mean deposition of cores are much smaller than the scatter of deposition values among the same eruptions in different cores (Fig. S4), we use the average deposition in our analysis, unless noted otherwise. This is supported by the abovementioned observation that even large eruptions can be entirely missing in individual cores, which underlines that the deposition values of individual cores are often not reliable.

Most of the eruptions in the data set of Lin et al. (2022) are not matched across Greenland and Antarctica. But the data set does include the bipolar eruptions previously identified by Svensson et al. (2020). Importantly, this dataset will be referred to as *unipolar* hereafter, even though the eruptions from Svensson et al. (2020) are still included.

## 2.2 Fine tuning and calibration of the eruption ages

The depths of the eruptions are not known with arbitrary precision, especially in ice cores where the underlying sulfur data sets are of low resolution and/or very noisy. Here we use the nominal depths reported in Lin et al. (2022) when investigating the unipolar data set, and the nominal depths from Svensson et al. (2020) when analyzing the bipolar eruptions. These depths are then transferred to the common age scale (see next Section), followed by a slight recalibration of the eruption ages, as explained in the following. First, there are slight systematic average offsets of the nominal depths compared to the sulfate maxima. This is a result of the detection of individual eruptions from noisy data combined with a slight asymmetry of the sulfate peaks, as well as the usage of multiple proxies in Svensson et al. (2020). In Fig. S1, the average sulfate peaks over bipolar and unipolar eruptions in all cores are shown, and one can see slight offsets of up to 2 years with respect to the nominal ages. Here we choose to correct these offsets and shift the eruption ages such that in each core the sulfate peaks on the age scales are aligned on average (see Sec. S1 for more details).

Second, we further shift the ages slightly by a fixed amount to account for the fact that the maximum sulfate peak in the ice core is delayed with respect to the eruption age. For large historic eruptions, comparable in size to the bipolar eruptions investigated here, this delay is estimated to be around 1.5 years (Burke et al., 2019). We shift all eruption ages back in time

by 1.5 years relative to the time of maximum sulfate deposition. Ideally, one would do this individually for each eruption by determining the start depth of the sulfate peak as an estimate of the actual starting year of the eruption. However, an individual age adjustment would only increase the jitter along the time axis, since the sulfate records are noisy due to intermittent de-

position and snow redistribution, and since the peaks of volcanic origin are subjected to smoothing by diffusion and different measurement techniques and resolution, which leads to peak widths that vary greatly across cores and time periods (Fig. S1 and S2). Thus, when interpreting our reported $\delta^{18}O$ anomalies averaged over different eruptions, it should be kept in mind that the events are aligned using the maximum sulfur deposition shifted by 1.5 years toward older ages, and not using the unknown, true time of the eruption start. In the plots where we report the time before eruption along the horizontal axis, the

135    year 0 indicates our estimate of the starting time of the eruptions as described here.

## 2.3   High-resolution oxygen isotopes

To quantify the climatic impact of the eruptions, we use high-resolution $\delta^{18}O$ records from 4 Greenland ice cores (NGRIP (NGRIP Members, 2004; Gkinis et al., 2014), GRIP (Johnsen et al., 1997), GISP2 (Stuiver and Grootes, 2000) and NEEM (Rasmussen et al., 2013)) on the annual layer-counted GICC05 time scale (Svensson et al., 2006, 2008; Rasmussen et al., 2013;

Seierstad et al., 2014), as well as from 3 Antarctic ice cores (WAIS (Buizert et al., 2015; Jones et al., 2018), EDC (Jouzel et al., 2007) and EDML (EPICA Community Members, 2006)) that have been matched to GICC05 at the bipolar volcanic eruptions (Svensson et al., 2020). All records cover the period from 11,700 years b2k (years before 2000 AD) to 60,000 years b2k. Since the records were measured at different depth resolutions and were taken at sites with different accumulation and thinning rates, their time resolution varies (see Tab. 1). The WAIS record was measured with continuous flow analysis (CFA), yielding

a data set with high depth resolution of 5mm, which results in the sub-annual time resolution given in Tab. 1. Note, however, that the effective measurement resolution is slightly lower due to mixing of material within the CFA apparatus (Jones et al., 2017). Perhgaps apart from the low-resolution EDML data set, in all records the true $\delta^{18}O$ time resolution, i.e., the degree of preservation of the temporal isotopic variability that was originally deposited on the ice sheet, is lower than given in Tab. 1. This is because of diffusion of the water molecules in firn and ice, which highly attenuates any variability below multi-annual

time scales for ice of the last glacial period, also in the high-resolution WAIS record (Cuffey and Steig, 1998; Jones et al., 2017, 2018).

For all time series used here, each data point consists of the average $\delta^{18}O$ value of bulk material in contiguous depth intervals. The data are thus not point samples, but averages over contiguous intervals. For our study the $\delta^{18}O$ records are processed in the following way. The midpoints of the depth intervals are interpolated linearly to the GICC05 time-depth scale,

yielding an unequally spaced time series. Next, this series is oversampled to a 1-year equidistant grid using nearest-neighbor interpolation. An exception is the sub-annual WAIS record, which is first averaged to 1-year resolution. Like this, the nature of the measurements as contiguous depth averages and the original measurement values are preserved, and all records are placed on the same equidistant time grid. We furthermore construct a stacked Greenland record in time slices around the bipolar volcanic eruptions. For a given eruption all individual cores where a depth has been recorded in Svensson et al. (2020) are

centered around the eruption depth and averaged.

For comparison with known historic eruptions we consider high-resolution Holocene $\delta^{18}O$ records from four different Greenland ice cores (NGRIP, GRIP, GISP2 and Dye-3 (Vinther et al., 2006)), covering the last 2,000 years. The time resolution in this period varies from monthly (Dye-3, GRIP) to biennial (GISP2). All 4 records have annual or higher resolution from 0-1.2 ka, and for the period >1.2 ka this is still the case for all cores except GISP2. The measured data on the depth scale is processed by interpolating the midpoints of the depth intervals linearly to the GICC05 time-depth scale, yielding an unequally spaced time series. Then, this series is oversampled to a monthly equidistant grid using nearest-neighbor interpolation. Only the Dye-3 record features a seasonal cycle, which is removed by a running yearly average. Subsequently, the records are stacked and a time series without trends and centennial or millennial variability is obtained via high-pass filtering the record by removing a Gaussian Kernel smoother with 150-year standard deviation.

**Table 1.** Median time resolution of $\delta^{18}O$ records (in years). The WAIS data was measured by a different technique, leading to a very high sample resolution. The true time resolution of the $\delta^{18}O$ signal is lower (see main text).

| Ice core | 11.7-20 ka | 20-30 ka | 30-40 ka | 40-50 ka | 50-60 ka |
|---|---|---|---|---|---|
| NGRIP | 1.8 | 2.8 | 2.6 | 2.8 | 3.0 |
| NEEM | 2.6 | 4.8 | 4.8 | 5.3 | 5.6 |
| GRIP | 4.6 | 4.0 | 4.4 | 4.8 | 5.2 |
| GISP2 | 4.3 | 11.3 | 12.7 | 13.8 | 15.4 |
| WAIS | 0.06 | 0.15 | 0.21 | 0.26 | 0.36 |
| EDC | 6.1 | 9.7 | 9.7 | 10.0 | 9.3 |
| EDML | 14.3 | 22.6 | 27.6 | 30.0 | 31.0 |

## 2.4 Records of layer-thickness

The NGRIP and WAIS cores have been layer-counted up to a certain depth. Subsequent depths of counted layers comprise an annual record of the layer thickness, which we use to study post-eruptive changes in accumulation rate. In NGRIP the layer-counting was performed until 60.2 ka BP, and thus the resulting record of annual layer thickness (Rasmussen et al., 2023) covers the entire investigated period. The counting includes certain and uncertain layers. For the certain layers, the depth increment corresponds to a one year time increment. In uncertain layers, which make up 10.1% of all layers, subsequent depths are defined as a half-year time increment (Andersen et al., 2006). To obtain the layer thickness record, we first convert the depth-age pairs of the GICC05 chronology to thickness-age pairs by taking the increments of subsequent depths. Then, to homogenize the record of full and half years, we linearly interpolate the record to a 0.1 year grid. The WAIS core was layer-counted until 31.2 ka BP (Sigl et al., 2016), thus only covering the younger part of the glacial. Here we use the layer-counted WD2014 chronology, which does not include half-years for uncertain layers. Otherwise, it is processed in the same way as for NGRIP.

## 3 Results

### 3.1 Volcanic cooling observed in Greenland after bipolar eruptions

We first consider the Greenland $\delta^{18}$O signal following the bipolar eruptions. For each eruption, the $\delta^{18}$O records are centered around the estimated time of eruption, and a segment of 50 years before and after the eruption is chosen and detrended linearly. To obtain anomalies we subtract the mean value of the detrended signal in the interval 10 to 50 years before the eruption. Finally, we extract the mean cooling anomaly from the non-volcanic variability by averaging these 100-year anomaly time slices over all eruptions. In Fig. 1a, the results are shown for the NGRIP core, which has the best temporal resolution. A negative multi-annual anomaly is seen, which clearly exceeds the variability in the mean signal leading up to the eruption. However, the mean anomaly is only approximately half the size of the high-frequency isotope variability around individual eruptions (gray bands). The other Greenland ice cores show the same qualitative behaviour, but the signals are less sharp due to the lower resolution (Fig. S5).

We attempt to remove non-climatic noise by averaging across all Greenland cores, as shown in Fig. 1b. Here the average isotopic cooling anomaly begins significantly prior to the estimated eruption age. This is due to diffusion of water molecules in firn and ice, as well as the averaging introduced by the isotope measurement on bulk material at multi-annual to decadal resolution. In addition, since the eruptions are aligned at the sulfate maxima and a constant 1.5 year shift towards older ages was applied to estimate the true eruption ages (see Sec. 2.2), for many eruptions we can expect the true age to be older than our estimate. This holds especially for larger eruptions with longer-lasting sulfate deposition, as well as for records with poor resolution and wider sulfate peaks (Fig. S1).

To quantify the isotopic cooling, we use the average signal (Fig. 1b) to define a time period corresponding to the most pronounced anomaly. This period consists of the estimated year of the eruption as well as the following five years (green shading in Fig. 1b). The average anomaly over this time period gives a scalar estimate of the volcanic cooling for each eruption, which we call the cooling amplitude hereafter. There is a rather weak correlation of this scalar cooling estimate of the individual eruptions among the Greenland cores (Fig. S6), suggesting large non-climatic noise in the high-resolution records. From the distribution of amplitudes in individual eruptions (Fig. 1c) it is clear that a non-negligible number of eruptions are followed by a positive $\delta^{18}$O anomaly, i.e., a potential warming associated with the eruption. For the Greenland stack, 23 out of the 82 bipolar eruptions feature a positive $\delta^{18}$O anomaly. It is unclear whether these eruptions indeed induced no volcanic cooling in Greenland, or whether it is masked by positive anomalies in the non-climatic noise and multi-annual climate variability. We construct a bootstrap distribution of 6-year anomalies of randomly chosen segments from the Greenland $\delta^{18}$O stack for the entire 12-60 ka period (gray distribution in Fig. 1c), which shows that natural fluctuations in $\delta^{18}$O anomalies of up to 1 permil are common. Using present-day calibrations of the $\delta^{18}$O thermometer of 0.69 permil/K to 0.8 permil/K (Sjolte et al., 2011; Buizert et al., 2014), this would correspond to 6-year mean temperature anomalies of up to 1.25 - 1.45 K, which would have the potential to mask the volcanic cooling of even very significant eruptions. Due to this large uncertainty, one cannot interpret the amplitude of individual eruptions as a quantitative estimate of the volcanic cooling. Nevertheless, the distribution

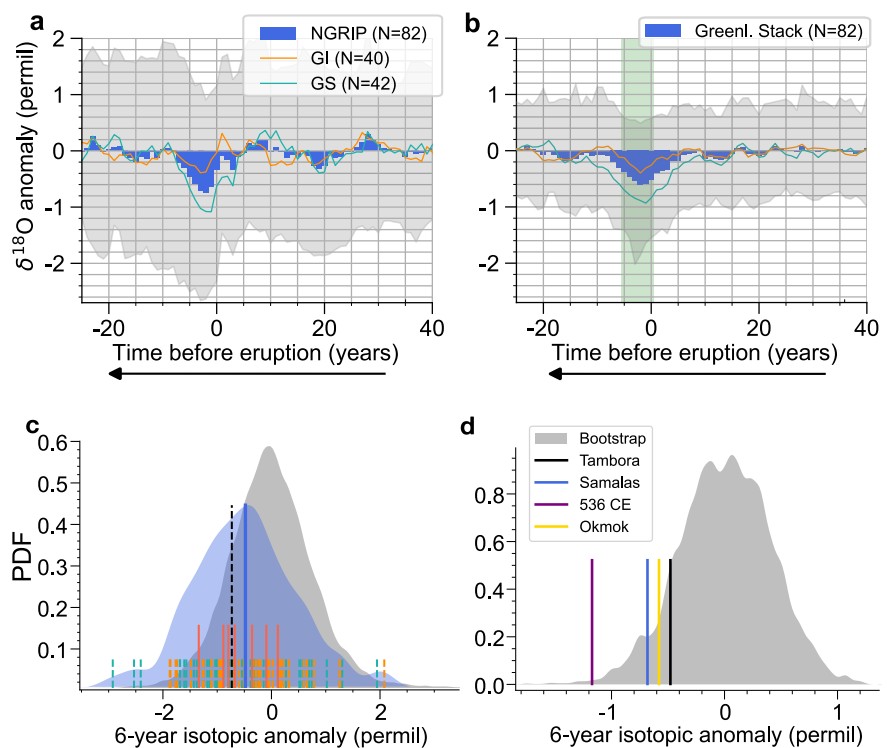

**Figure 1. a** Average NGRIP $\delta^{18}$O anomaly centered at the bipolar eruptions, defined with respect to the mean of the period 10-50 years prior to the eruption. The average signal is shown in blue, and the gray bands are the 16- to 84-percentiles of detrended time slices covering individual eruptions. In orange (green) we show the mean signal of eruptions during GI (GS). **b** Same for the Greenland stack, where detrended slices of all cores for every eruption are averaged, using only cores where a depth is identified (49 eruptions with four cores, 14 (10) with three (two) cores, and 9 represented by NGRIP only). **c** Distribution of 6-year average anomalies of the Greenland $\delta^{18}$O stack around the eruptions (blue), compared to a bootstrap of randomly chosen 6-year anomalies from the stack using all 4 cores on the GICC05 synchronization (gray). The black dashed line is the 16-percentile of the bootstrap distribution, and the blue line is the mean of the volcanic anomalies. Dashed orange (green) lines show individual eruptions during GI (GS). Red lines are eruptions preceding the onsets of Dansgaard-Oeschger events within less than 50 years, as identified in Lohmann and Svensson (2022). **d** Anomalies of the Holocene Greenland stack (see Methods). Shown is the cooling amplitude of several major Common Era eruptions, as well as the bootstrap distribution of random segments from the past 2 kyr. The historic eruptions are 1815 CE Tambora, 1258 CE Samalas, and 43 BCE Okmok, as well as the 536/540 CE doublet. For the latter we chose the age of 536. In most $\delta^{18}$O records the doublet is merged due to diffusion. GICC05 ages are taken from McConnell et al. (2020) and Sigl et al. (2015).

of amplitudes is clearly shifted towards negative values, unlike the bootstrap distribution, which is symmetric and centered at 0.

We define a signal-to-noise ratio (SNR) of the volcanic signal by dividing the mean volcanic anomaly (blue line in Fig. 1c) by the 16-percentile of the bootstrap distribution (as a measure of standard deviation, black dashed line in Fig. 1c). This is not

a SNR of the record as a temperature proxy in absolute terms. We consider only the volcanic cooling anomaly as signal, while internal temperature variability not caused by volcanism is considered noise, alongside actual proxy noise from intermittent precipitation and post-depositional processes. Thus, what we define as SNR measures the strength of the multi-annual volcanic cooling signal relative to the multi-annual climatic and non-climatic proxy variability. For the Greenland stack this yields $SNR = 0.66$, and for NGRIP we find $SNR = 0.48$. Thus, stacking improves the SNR, but the average anomaly still does not exceed the non-volcanic variability. While noise in the vertical axis of Fig. 1b is reduced when stacking different cores, additional noise is introduced in the horizontal axis since not all cores have an equally good alignment of the isotope record relative to the true eruption age. This is because the precise eruption depth is less certain in some cores due to low resolution of the underlying sulfate records (Fig.'s S1 and S2). Further, there are small systematic offsets in the depth scale of $\delta^{18}O$ and sulfate measurements of the same ice core, as they are not obtained from the same samples.

The average 6-year cooling amplitude is 0.48 permil in the stack and 0.63 permil in NGRIP. This may be compared to the largest eruptions in the Common Era. These are much better constrained since most of them have an identified source, a well quantified magnitude, as well as a precise date, allowing them to be matched to other paleoclimate proxies, such as from tree rings (Sigl et al., 2015). In Fig. 1d we show a distribution of anomalies from randomly chosen segments of a Greenland $\delta^{18}O$ stack covering the last 2,000 years (Methods), together with the cooling amplitude of four major historic eruptions that have been estimated to be among the 5 largest eruptions during this time interval (Sigl et al., 2015). These feature an average negative $\delta^{18}O$ anomaly of 0.73 permil. The average isotopic anomaly of the bipolar eruptions during the glacial is thus slightly weaker by comparison. However, the calculated glacial $\delta^{18}O$ anomalies are likely underestimated compared to the Common Era eruptions due to several factors discussed in Sec. 3.3.

### 3.2 Volcanic cooling observed in Antarctica after bipolar eruptions

The average volcanic isotopic anomaly in Antarctica is more subdued, which may be expected as Antarctica is climatically relatively isolated and more volcanos are located in the Northern Hemisphere. The WAIS record is a priori best suited to show a clear volcanic cooling signal due to its high accumulation rate and measurement resolution. A roughly 4 year long average negative $\delta^{18}O$ anomaly is found (Fig. 2a), but it is only marginally significant as it is not much larger than the variations in the mean anomaly before and after the eruption. The average $\delta^{18}O$ cooling anomaly in EDC is much less sharp (Fig. 2b), because of the low accumulation rate. This yields an average $\delta^{18}O$ resolution of almost 10 years, along with pronounced diffusion and non-climatic noise (Münch et al., 2016). The EDML core does not show any cooling signal for the bipolar data set (Fig. S5d), partly because its isotopic resolution is too low. In addition, $\delta^{18}O$ records close to coastal regions have been found to capture only very little local temperature variability on short time scales (Vega et al., 2016; Goursaud et al., 2019). Nevertheless, by averaging over many eruptions from the unipolar data set a slight cooling anomaly can be discerned (not shown here).

We again define a period of most pronounced cooling based on the average anomaly curves. For WAIS this corresponds to the estimated eruption year, as well as the year before and the two after. Figure 2c shows that the cooling amplitudes associated with bipolar eruptions, which average at -0.20 permil, are only shifted slightly towards negative values compared to randomly selected periods of the record. For EDC we choose an almost symmetric period with 7 years before and 6 years after the

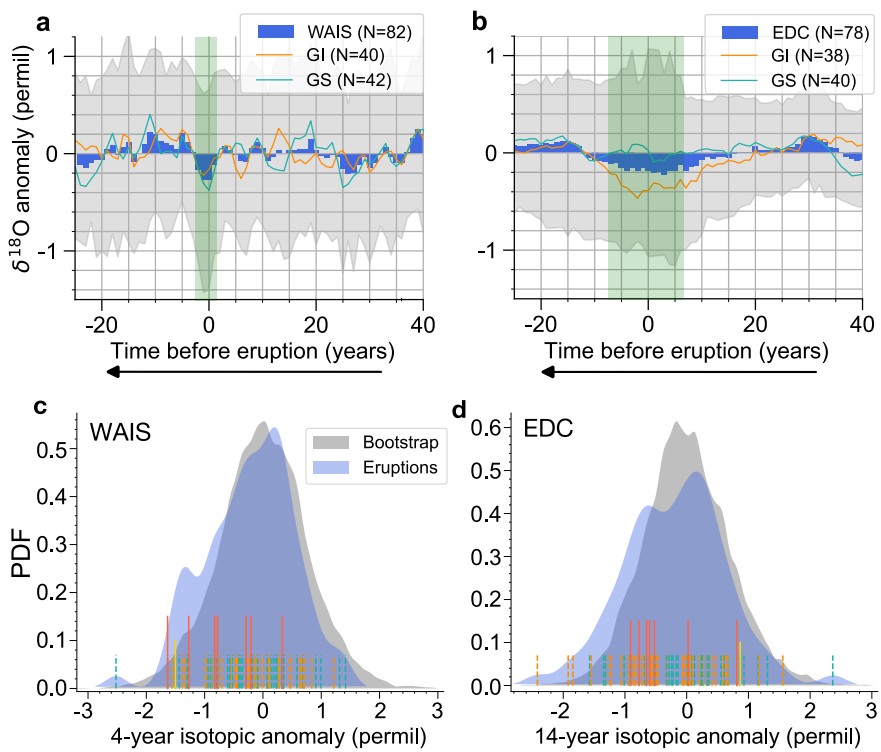

**Figure 2.** Same as Fig. 1b-c, but for the Antarctic cores WAIS (**a,c**) and EDC (**b,d**). For EDC, only 78 out of 82 bipolar eruptions were detected in Svensson et al. (2020).

estimated eruption year. This also yields an average anomaly of -0.20 permil and a slight negative shift of the distribution of individual anomalies (Fig. 2d). Since the EDC record has a much lower sample resolution and thus more pronounced smoothing due to averaging, the original peak anomaly would be clearly larger in absolute terms compared to WAIS. Still, compared to the proxy background variability, the average volcanic signal is similar for the two cores. The SNR derived from the distributions in Fig. 2c,d is $SNR = 0.30$ for WAIS and $SNR = 0.28$ for EDC. These low values highlight that the volcanic cooling signals are weaker in the Antarctic cores compared to Greenland, which may be in part due to a more muted or variable Antarctic climate response, but also due to poorer performance of the $\delta^{18}O$ proxy. Indeed, cooling amplitudes of individual eruptions in WAIS and EDC are not significantly correlated, and the amplitudes of both Antarctic cores are also not correlated with the amplitudes from the Greenland stack (Fig. S7).

### 3.3 Preservation of the cooling signal in the isotope record

The above estimates of the average multi-annual isotopic cooling anomaly are a compound of young eruptions and older ones, for which the signal is degraded due to several effects. First, multi-annual $\delta^{18}O$ anomalies are smoothed out by diffusion of water molecules in the ice. The older the ice the more time has elapsed for the diffusion to act. Additionally, deeper annual

layers become thinner due to ice flow, which leads to increasing ice diffusion length (in years) with depth (and thus age). Second, there is additional smoothing due to the measurement of $\delta^{18}O$ on contiguous pieces of the ice core at constant depth intervals. For thinning annual layers with age, this smoothing by averaging is more pronounced the older the eruption. Third, the effective temporal resolution of the underlying sulfate records typically decreases with age (Tab. S1). As a result the eruption age can be determined less accurately for older eruptions (Fig. S2a), which again leads to a smearing out of the average cooling anomaly.

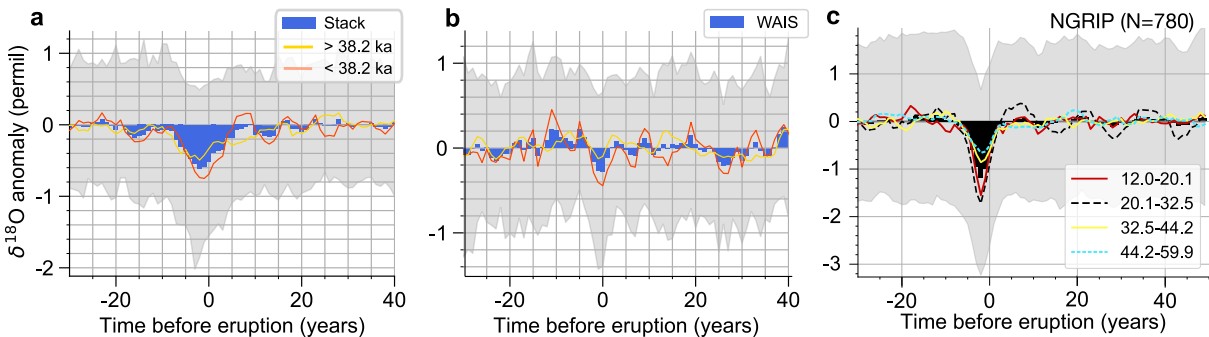

**Figure 3. a,b** Average $\delta^{18}O$ anomalies in the Greenland stack (**a**) and the WAIS record (**b**) aligned to the bipolar eruptions. The gray shading and blue curve are the same as in Fig's. 1b and 2a. The red (yellow) curves correspond to the average $\delta^{18}O$ anomaly of the younger (older) half of the bipolar data set. **c** Average $\delta^{18}O$ anomalies in the NGRIP record aligned to the unipolar eruptions. The average signal is shown in black, and the gray bands are the 16- to 84-percentiles of detrended time slices covering individual eruptions. The colored curves are the average signals of four equal sized subsets of eruptions divided according to age.

Consequently, while the average magnitude of the eruptions measured by their sulfate deposition does not appear to change over the course of the glacial (see Fig. S8 and S9, as well as Lin et al. (2022)), younger eruptions show a more pronounced cooling anomaly compared to older ones (Fig. 3a,b, and Fig. S10 for all other cores). In the Greenland stack, the younger half of eruptions show a minimum anomaly of -0.75 permil in the year after the eruption. With the abovementioned present-day calibrations of the $\delta^{18}O$ thermometer of 0.69 permil/K to 0.8 permil/K (Sjolte et al., 2011; Buizert et al., 2014), this yields a peak cooling of 0.94-1.09 K, which comes close to the 1.24 K summer NH cooling estimated from tree rings for the largest 4 eruptions of the Common Era (Sigl et al., 2015).

The evolution over time of the $\delta^{18}O$ cooling anomaly can be investigated more precisely using the NGRIP core in isolation, which has the highest $\delta^{18}O$ and sulfate resolution, as well as the best dating. Here, in the younger half of the bipolar eruptions the 6-year mean isotope amplitude is -0.77 permil and the peak cooling amplitude is -0.90 permil. Using the unipolar eruption record, which features many more eruptions and thus a less noisy mean signal, we find that eruptions occurring in the period 12-32 ka feature a minimum anomaly of -1.7 permil two years after the estimated eruption age (Fig. 3c). Thus, despite a return period of only 65 years, the cooling anomaly of the youngest glacial eruptions clearly exceeds the anomaly after the largest eruptions in the Common Era. For the older eruptions, the minimum anomaly is attenuated by roughly a factor of 2.

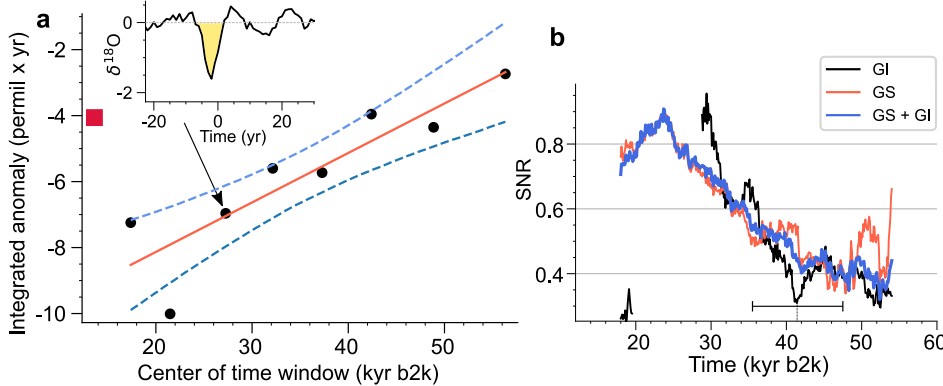

**Figure 4. a** Average integrated $\delta^{18}O$ anomaly in NGRIP for the unipolar data set, where the eruptions are separated according to age in consecutive bins with an equal number of eruptions (10 bins with 78 eruptions each). The $\delta^{18}O$ time slice around each eruption in a bin is detrended as described in the main text. The detrended slices of eruptions in a bin are then averaged to yield the mean $\delta^{18}O$ anomaly, from which the integrated anomaly is defined as the area above the consistently negative signal around the time of the eruption (yellow shading in the inset). The red square corresponds to the value for the youngest bin of eruptions, occurring from 15,685 to 12,022 years BP, and it is a thus far unexplained deviation from the roughly linear decrease of the anomaly over time, as indicated by the linear fit. **b** Signal-to-noise ratio in the NGRIP record estimated in a 12 kyr moving window. The method, explained in Sec. 3.1, is applied here to the unipolar dataset using 4-year average anomalies starting with the year of the eruptions. Shown are curves for all eruptions, as well as for the GI and GS subsets. The GI curve is interrupted from 20-28 ka, as there are too few eruptions (less than 20 per 12 kyr) for a robust SNR estimation.

The amplitude of the isotopic cooling signal at the time of deposition is expected to be even larger, because a) the $\delta^{18}O$ records are not perfectly aligned to the true eruption year, and b) the smoothing effect of diffusion has not been accounted for in our study. There are techniques to achieve the latter, if the diffusion length in ice and firn is known (Johnsen et al., 2000). Here we refrain from doing so, because the variations over time of the cooling anomalies do not appear to follow a simple diffusion process. While the peak cooling anomalies in Fig. 3c decrease over time, the anomaly does not get visibly smeared out further in time. The area under the curve corresponding to the negative anomalies does not stay constant, as expected for a simple diffusion of temperature fluctuations over time, but decreases over time (Fig. 4a). Further, in contrast to a constant SNR due to a roughly equal diffusive attenuation of the noise background and volcanic signal, the SNR decreases over time (Fig. 4b). This may reflect the additional attenuation effect on the volcanic signal of the decreasing precision of the eruption alignment going further back in time.

### 3.4 Correlation of cooling signal to volcanic magnitude and hemispheric sulfur deposition

We next investigate whether eruptions that were large in terms of their sulfate deposition also led to a more pronounced isotopic cooling. Considering the total sulfur aerosol loading, which is a combined metric of the Greenland and Antarctic sulfate deposition (Lin et al., 2022), the bipolar data set is divided into two, depending on whether an eruption was larger or smaller compared to the median. The larger eruptions feature a much more pronounced $\delta^{18}O$ anomaly, as shown for the

Greenland stack and EDC in Fig. 5a,b. For individual eruptions there is only a weak correlation of $\delta^{18}$O anomaly and aerosol loading (Fig. S11). This is because of the high noise levels in the records, which, together with the relatively small size of the bipolar sample, limits our ability to quantify the dependence of the $\delta^{18}$O anomaly on the sulfate deposition. For this purpose the larger unipolar dataset is used in the following, with the caveat that only the unipolar deposition is available, and not the total aerosol loading.

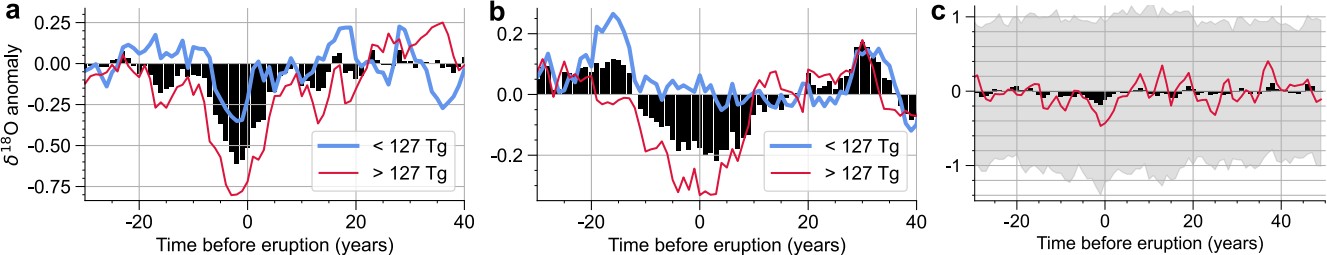

**Figure 5. a,b** Average $\delta^{18}$O anomalies in the Greenland stack (**a**) and the EDC record (**b**) aligned to the bipolar eruptions. The black curve is the average signal over all bipolar eruptions, and the red (blue) curves correspond to the average $\delta^{18}$O anomaly of half the bipolar data set with a larger (smaller) total sulfur aerosol loading (Lin et al., 2022) compared to the median (127 Tg). **c** Mean $\delta^{18}$O anomalies (permil) in the WAIS core in response to the unipolar eruptions. Black bars show the mean signal, and the gray bands are the 16- to 84-percentiles of detrended time slices covering individual eruptions. The red curve corresponds to the mean anomaly in the largest 10 percent of eruptions, in terms of their WAIS sulfate deposition magnitude.

Also in the unipolar data set eruptions with a larger sulfate deposition show a more pronounced average $\delta^{18}$O anomaly. The WAIS core, for instance, which showed only a weak average anomaly after bipolar eruptions, features a much stronger cooling signal for the eruptions with largest unipolar sulfate deposition (Fig. 5c). We next define a scalar metric for the cooling associated with eruptions of a certain size category. To this end we separate the eruptions from the unipolar data set of a given core into bins according to their unipolar sulfate deposition. Here we consider the two cases: a) the deposition values are only taken from one core; or b) the deposition values are averaged over all cores where a given eruption was identified (see Sec. 2.1). We then calculate the average $\delta^{18}$O anomalies of all eruptions in one bin, and integrate the resulting mean signal over the years around the eruption where a negative anomaly is found (yellow area in inset of Fig. 6a, see also Fig. S12). For all cores, this integrated isotopic anomaly shows a clear relation with the unipolar deposition magnitude (Fig. 6a and Fig. S13). A linear fit to the data seems justified in most cases, but we cannot rule out a non-linear relation. For most cores the linear fit indicates that there is still a significant negative isotopic cooling anomalies when extrapolating to zero sulfate deposition. We speculate this could be because a) the linear relationship breaks down for the smallest eruptions that still have a global cooling effect but no polar sulfate deposition, or because b) the largest sulfate deposition values are inflated due to a significant proportion of local or regional eruptions with a large tropospheric sulfate transport and polar deposition. There is generally a better correlation of the anomaly with the deposition in the individual core, and not the deposition averaged over multiple cores (Fig. S13). This may be surprising since the latter should be a more reliable estimate for the magnitude (Sec. 2.1). A reason for this may be that

the eruptions with a large depositional sulfate peak in the respective core feature a more precise depth estimate, leading to a better average alignment of the isotopic cooling anomaly to the true age of the eruption.

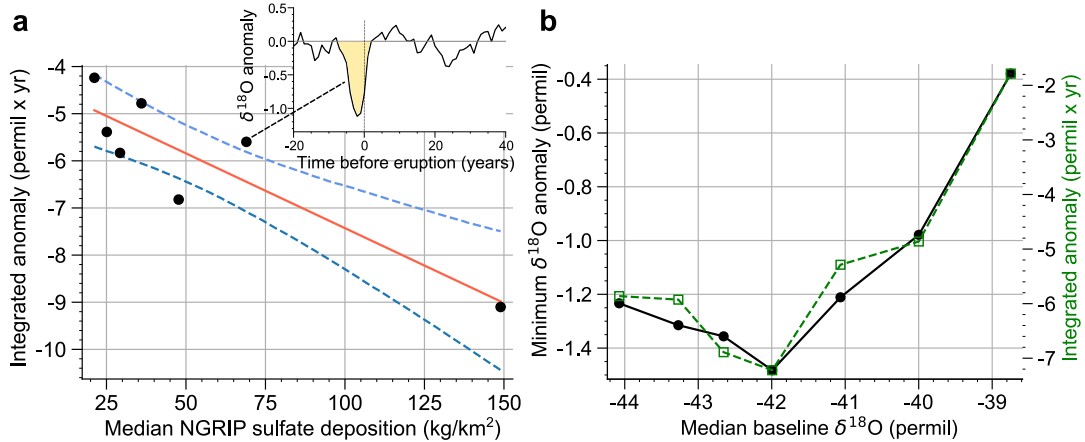

**Figure 6. a** Correlation of the integrated NGRIP $\delta^{18}$O anomalies (eruptions from the unipolar dataset) and the associated sulfur deposition in the NGRIP core (Lin et al., 2022). Individual dots represent the average integrated anomaly of the eruptions divided into bins according to the 15-, 30-, 45-, 60-, 75-, and 90-percentiles of the NGRIP sulfur deposition. The integrated anomaly is defined by the sum of averaged $\delta^{18}$O anomaly values in those years around the estimated year of eruption where the anomaly is negative, as shown with the shaded area in the inset. We also show a linear regression with 95% confidence intervals, which has a non-zero intercept of -4.3 permil. **b** Average NGRIP $\delta^{18}$O anomaly as a function of the baseline $\delta^{18}$O level, defined by the mean $\delta^{18}$O value from 50 up until 3 years before the unipolar eruptions. The data is averaged in equally sized bins according to the $\delta^{18}$O baseline values. Shown is the minimum of the average $\delta^{18}$O anomaly (black), as well as the integrated average $\delta^{18}$O anomaly, as was used in panel **a** (green dashed).

Based on the relative deposition of bipolar eruptions in Greenland and Antarctica, the source latitudes have been classified in
binary categories with Northern Hemisphere (NH above 40 deg N) or Southern Hemisphere and low latitude (SH/LL) eruptions (Lin et al., 2022). Since there is a correlation of the isotopic anomaly with the unipolar deposition magnitude in all cores, we see an according stronger Greenland (Antarctic) isotopic response for NH (SH/LL) eruptions (Fig. 7). For eruptions with a larger Greenland sulfate deposition (classified as NH) there is no significant EDC $\delta^{18}$O cooling anomaly. It may be that a non-negligible number of these eruption even feature a warming anomaly, which might be reflected in the positive $\delta^{18}$O excursion in
the confidence bands for the lower resolution Antarctic cores (Fig. 2b, S5d and Fig. 7b), and in the slight indication of a bimodal distribution in Fig. 2d. Note, however, that a certain widening of the confidence bands is expected in the low resolution records due to the detrending and nudging of the anomaly to the period prior to the eruption. Moreover, since there are relatively more NH classified eruptions in GS compared to GI, we cannot clearly separate the effect of the estimated eruption latitude on the $\delta^{18}$O anomaly from the even more pronounced GI-GS contrast (see next Section). Larger bipolar data sets would be required
to resolve this, and at this stage we believe that neither the determination of the eruption latitude and the inferred volcanic cooling from the $\delta^{18}$O proxy are precise enough to warrant much speculation on the dependence of the climate response as a function of the eruption site.

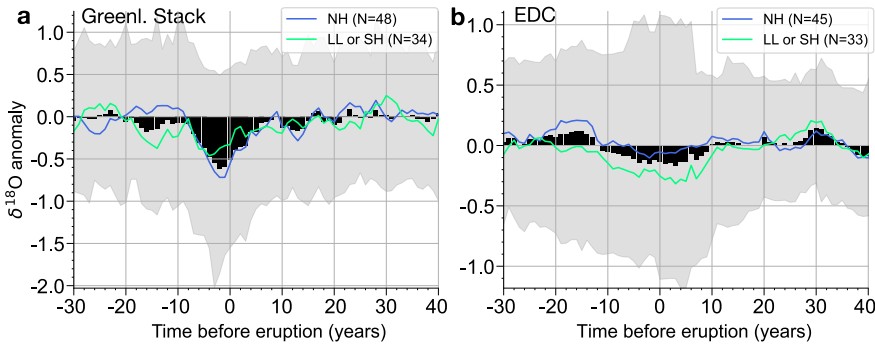

**Figure 7.** Mean $\delta^{18}$O anomalies (permil) in response to the bipolar eruptions in the Greenland stack (**a**) and EDC (**b**) records. The black curve and gray shading are as in Fig.'s 1b and 2b, and the colored curves are the average anomalies of the subsets of eruptions classified by Lin et al. (2022) as Northern Hemisphere (NH, above 40deg North) and low-latitude/Southern Hemisphere (LL/SH, below 40deg North), based on the relative sulfur deposition measured in Greenland versus Antarctica.

## 3.5    State-dependency of the climate response

In Sec. 3.3 we found that the younger, best preserved glacial eruptions in NGRIP feature a significantly stronger isotopic cooling
compared to the largest eruptions during the Common Era. This indicates a state dependency of the volcanic $\delta^{18}$O anomaly, which could reflect a state dependency of climate sensitivity, i.e., of the response of the global average surface temperature to the radiative forcing of the volcanic eruption. Instead of global climate sensitivity, there could also be a state dependency in regional climate sensitivity, i.e., a difference in the spatial pattern of the temperature change. Here we use the term climate sensitivity to describe the temporary temperature change following the relatively short-lived volcanic perturbation. This differs from the
concept of equilibrium climate sensitivity, which describes the long-term increase in global temperature as a response to an instantaneous, permanent doubling of atmospheric $CO_2$. The response to short-term volcanic forcing involves fast components of the climate sensitivity, but longer-term climate feedbacks and changes in deep ocean heat content are limited. Accordingly, modeling studies have found difficulties in using observations of volcanic cooling to constrain equilibrium climate sensitivity (Boer et al., 2007; Bender et al., 2010; Merlis et al., 2014).
It is also possible that the temperature response is essentially independent of the background climate state, and that instead there is a state dependency of the proxy sensitivity, i.e., of the magnitude of $\delta^{18}$O change for a given volcanic temperature change. Compared to the present-day proxy sensitivity, previous work suggests that the $\delta^{18}$O proxy in Greenland reacts more sensitive to the temperature changes across glacial regime shifts, such as the last deglaciation and DO events (Guillevic et al., 2013; Buizert et al., 2014), while the opposite is the case for Antarctic $\delta^{18}$O (Uemara et al., 2012; Buizert et al., 2021). This
is due to a combination of changes in accumulation seasonality, moisture source, and ice sheet topography associated with the regime shifts. But the sensitivity of the proxy to short-term temperature changes without major regime shifts is unknown, and its dependence on the background climate state remains an active subject of research (Liu et al., 2023; Cauquoin et al., 2023). Comparing the mean volcanic $\delta^{18}$O anomaly to the baseline $\delta^{18}$O values at which the corresponding eruptions occurred, we

find a non-linear dependence of the anomaly on the background state (Fig. 6b). This could be interpreted as a state dependency of the proxy or climate response, but it partly reflects the better signal preservation for the predominantly young eruptions occurring at low $\delta^{18}$O baseline values, as a result of the gradual decrease of $\delta^{18}$O values throughout the glacial.

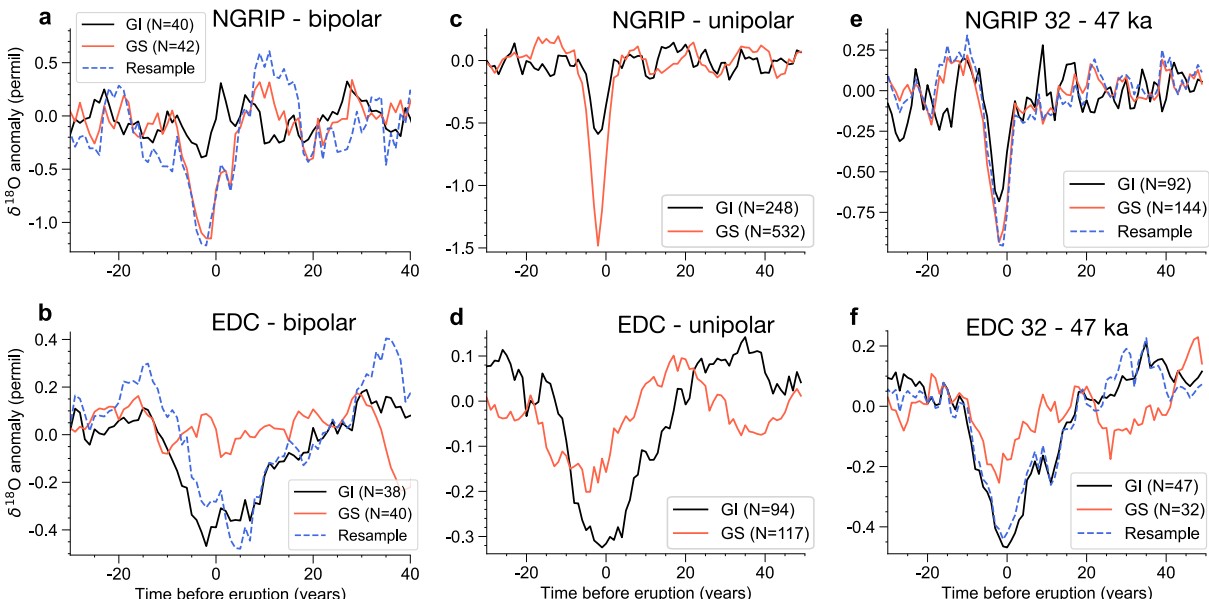

**Figure 8.** Average $\delta^{18}$O anomaly in the NGRIP (top panels) and EDC (bottom panels) core, obtained by aligning the records at the volcanic eruptions from the (**a,b**) bipolar and (**c,d**) unipolar data sets, as well as a subset of the unipolar data set representing a time period with an equal proportion of GS and GI conditions (**e,f**). The eruptions are separated into subsets occurring during GI (GS), and the average $\delta^{18}$O anomaly is shown in black (red). The blue dashed lines show the anomaly curves obtained when resampling the GS subset in NGRIP and GI subset in EDC, such that the distribution of the associated magnitudes of sulfur deposition matches the distribution of the eruptions in the corresponding GI subset for NGRIP and GS subset for EDC (see main text for more detail).

A more conclusive picture of the state dependency for both Greenland and Antarctica can be obtained by dividing the data sets in eruptions occurring during the cold (GS) and mild (GI) periods of DO cycles. While changes in Antarctic climate over DO cycles are much weaker compared to Greenland, the DO cycles are nevertheless the most pronounced large-scale climate regime shifts of the last glacial. Thus, dividing the data into GI and GS periods indicates which part of the DO cycle the global climate state occupies. This seems a reasonable target to test the climate state dependency for both Greenland and Antarctica. Eruptions occurring during GS show a more pronounced isotope anomaly in Greenland compared to eruptions during GI, while the opposite is the case for the Antarctic EDC core (coloured lines in Fig. 1a,b and Fig. 2b). The response pattern in WAIS seems similar to Greenland (Fig. 2a), but it is inconclusive since the signals are not larger than the variability before the eruptions. Figure 8a,b shows the mean anomaly signals in NGRIP and EDC in more detail. The stronger Greenland GS response is surprising, because we would a priori expect a sharper volcanic cooling response in GI due to the higher

accumulation rate resulting in a higher resolution and less pronounced non-climatic noise. Further, the higher accumulation rate also leads to a higher-resolution sulfate record, and thus a sharper estimate of the eruption depth.

Using the much larger unipolar data sets, the difference in response is also seen clearly (Fig. 8c,d). However, this data set (unlike the bipolar one) contains the last glacial maximum, which features almost exclusively stadial conditions, and which occurs during the younger part of the glacial where the signal preservation is better (Sec. 3.3). For a more fair comparison, we choose the interval 32-47.5 ka in the middle of our time period, which features an equal number of years with GI and GS conditions (Fig. S14). Even though reduced in NGRIP, the contrasting isotopic response is still significant in this interval (Fig. 8e,f). This difference in GI versus GS could be due to several factors:

1. There is a different global or regional climate sensitivity to identical radiative forcing

2. The effective radiative forcing of (identical) sulfur-rich eruptions is different

3. The global or regional volcanic activity was different in GS versus GI.

4. The dependence of $\delta^{18}$O on annual mean surface temperature in Greenland and Antarctica varied for GS and GI.

5. The influence of factors other than annual mean temperature on $\delta^{18}$O anomalies is different in GS and GI.

Since the SNR in GI and GS is similar for most parts of the record (Fig. 4b), the increase in inferred volcanic cooling during GS compared to GI equals the increase in non-volcanic proxy variability, which is consistent with a state dependency of both climate and proxy sensitivity. A state dependence of climate sensitivity (point 1.) would be an intriguing finding, but it is hard to rule out the confounding factors (points 2.-5.). Since the state dependency of the $\delta^{18}$O anomaly is opposite in Greenland and Antarctica, the potential state dependency is more likely in the regional climate sensitivity, i.e., the spatial pattern of the volcanic cooling, rather than in global average temperature. In the next section, we analyze differences in the volcanic forcing between GS and GI (point 3. and to some extent 2.). In the section thereafter we employ records of relative snow accumulation rate in an attempt to gather more evidence for state dependency of the $\delta^{18}$O-temperature relationship (point 4.), as well as for influences of relative accumulation rate changes on the $\delta^{18}$O signal (as part of point 5.).

### 3.6 State-dependency of volcanic forcing

There is generally a higher frequency of eruptions detected in Greenland during GS (Fig. 9a and see also Lin et al. (2022)). To some degree, this may be an artifact of the automatic detection of eruptions, because the estimated eruption magnitudes could depend on the background noise level in the sulfate records, which is very different in GS and GI (Lin et al., 2022). But for the average climate impact of eruptions only the relative distribution of the magnitudes counts, and not the absolute frequency of the eruptions. The distribution of sulfate deposition in Greenland is skewed towards larger values during GS (Fig. 9b,c), whereas in Antarctica the distribution of GI eruptions is skewed to larger values (Fig. 9d,e).

This gives a consistent pattern with larger eruptions and more pronounced isotopic cooling in Greenland during GS, and conversely larger eruptions and more cooling during GI in Antarctica. But in the following we show by resampling that the

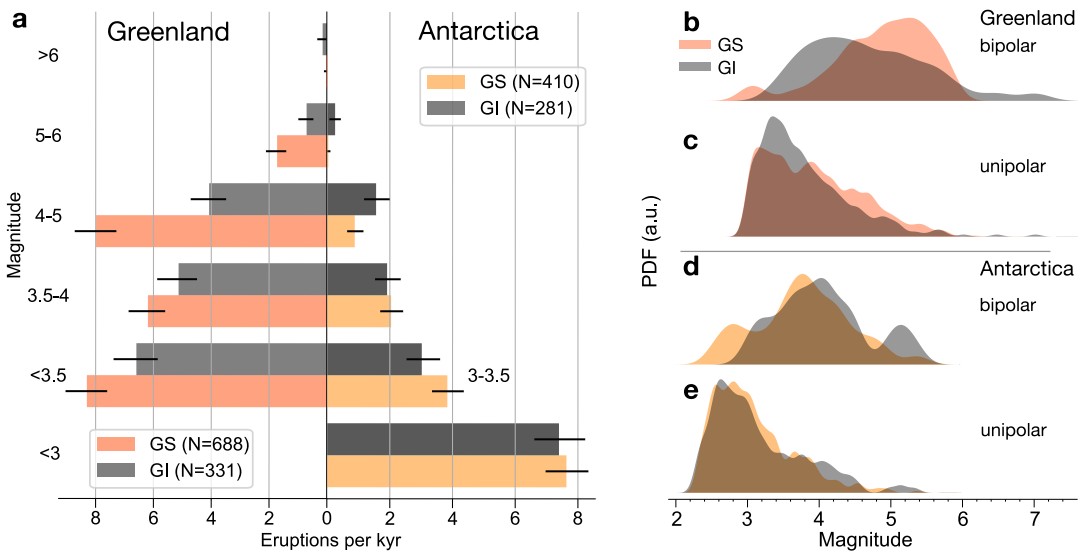

**Figure 9. a** Frequency of eruptions in the Greenland and Antarctic unipolar data sets in different magnitude categories (defined as the logarithm of the unipolar sulfate deposition in kg/km$^2$), which was derived as averages over the deposition in the individual cores where the eruptions could be detected (Lin et al., 2022). The data sets are further split up in eruptions occurring during GS and GI. The error bars on the frequency estimates are given as black lines, and represent the 10- to 90-percentile computed analytically assuming a Poisson distribution for the number of eruptions occurring in the respective time intervals. **b-e** Distributions of the magnitude of the eruptions, given by the logarithm of the unipolar sulfate deposition, for the Greenland (**b-c**) and Antarctic (**d-e**) data sets, which are divided into eruptions occurring during GS and GI.

differences in sulfur deposition magnitude cannot explain the contrasting $\delta^{18}O$ response. In particular, we resample the subset of eruptions with a larger average deposition (i.e. the GS eruptions for Greenland and the GI eruptions for Antarctica) with replacement such that they match the deposition magnitude distribution of the other subset with lower average deposition. From this resampled set of eruptions we calculate the average $\delta^{18}O$ response and compare it to the subsets before resampling. The resampling method is explained in the Appendix of Lohmann and Svensson (2022), and it is similar to established Monte Carlo methods such as importance sampling, which aim to generate samples from a particular distribution, when only having observed samples from another distribution. The sulfur deposition distributions before and after resampling are shown in Fig. S15, and the resulting resampled average $\delta^{18}O$ anomaly is shown by the dashed lines in Fig. 8a,b and Fig. 8e,f. In Greenland, the average isotopic anomaly is still more pronounced for the GS eruptions, and the same holds true for the GI eruptions in Antarctica. The results also hold for the other Greenland cores (see Fig. S16 for NEEM). Thus, the contrasting isotopic response in GI versus GS cannot be explained by the observed differences in the distribution of sulfate depositions. It may be that the latter is due to differences in sulfur transport and not the amount of sulfur ejected. There could be GI-GS differences in wind speeds and circulation patterns, which may or may not influence the aerosol climate forcing. Differences in atmospheric moisture may also modulate the lifetime and climate forcing of sulfur aerosols. A longer sulfate lifetime in the dryer GS may be visible in

broader Greenland sulfate peaks (Fig. S2c), but we cannot distinguish this from a broadening of the peaks due to the lower resolution during GS.

## 3.7 State-dependent volcanic impact on accumulation rate

Due to the unknown and potentially varying sensitivity $\alpha$ of the $\delta^{18}O$ proxy to temperature changes, the implied state dependency of the volcanic cooling may be spurious. To get additional evidence, we reconstruct changes in precipitation after the eruptions. Precipitation changes are expected to follow radiatively induced changes in temperature, since the atmospheric moisture capacity varies exponentially with temperature (Clausius-Clapyeron relation, CC). Indeed, volcanic cooling leads to a reduction in precipitation due to a weakened hydrological cycle (Robock and Liu, 1994; Bala et al., 2008). In the polar

regions, short-term relative changes in snow accumulation rates $\lambda$ can be almost directly monitored in layer-counted ice cores by comparing the average layer thickness (implied by the depths of the counted annual layers) of close-by time intervals. Unlike $\delta^{18}O$, this is a direct measurement and its annual resolution is only slightly blurred by the imprecisions of the layer identification. We follow CC by assuming a change of $\lambda$ with temperature

$$\frac{\partial \lambda}{\partial T} = \gamma \lambda, \tag{1}$$

and obtain $\lambda \propto e^{\gamma T}$, where $\gamma$ is the accumulation sensitivity. Thus, the logarithm of the ratio of $\lambda$ before and after a temperature change $\Delta T = T - T_0$ is linearly related to $\Delta T$ and by extension to the measured $\delta^{18}O$ change for a given isotope sensitivity $\alpha$:

$$\Delta \log \lambda \equiv \log \frac{\lambda(T)}{\lambda(T_0)} = \gamma \Delta T = \frac{\gamma}{\alpha} \Delta \delta^{18}O. \tag{2}$$

$\gamma_{CC} = 0.073$ would be found when deriving Eq. 1 from a linearized Clausius-Clapeyron equation, assuming that the total

435 precipitation amount is proportional to water vapor pressure. But the true value of $\gamma$ for the climate system is lower, and varies with location and $T_0$ (Allen and Ingram, 2002).

We now consider anomalies with respect to the average state ($T_0$, $\lambda(T_0) \equiv \lambda_0$) during the 50 years prior to the eruption. Figure 10a,b shows the percentage change anomalies of $\lambda$, defined as $(\frac{\lambda}{\lambda_0} - 1) \cdot 100$, for the NGRIP and WAIS unipolar data sets. Indeed, there are reduced accumulation rates in NGRIP associated with the eruptions in both GI and GS. The reduction is

440 clearly more pronounced in GI. For WAIS, while the reduction in GS is not significant compared to the variability of the mean, there seems to be a more pronounced reduction in GI. However, the signal-to-noise ratio is low because the layer thickness is strongly affected by surface snow redistribution, and thus an average over a large number of eruptions is needed to extract the signal (compare mean signal and gray band in Fig. 10c). The seemingly delayed peak reduction in WAIS for GI eruptions might thus only be a random feature of the variability in the mean due to the small sample size of GI eruptions.

Focusing on NGRIP, the maximum layer thickness change averaged over all eruptions is 5.6±0.9%. The error is estimated by the standard deviation of the mean before the eruptions (fluctuations of the mean curve in Fig. 10c). This is larger than the 3% global precipitation reduction inferred via sea level changes after 5 major eruptions during the last century (Grinsted et al., 2007), or the modeled reductions of 1-2% for the same eruptions (Iles and Hegerl, 2014). This could be due to an amplified

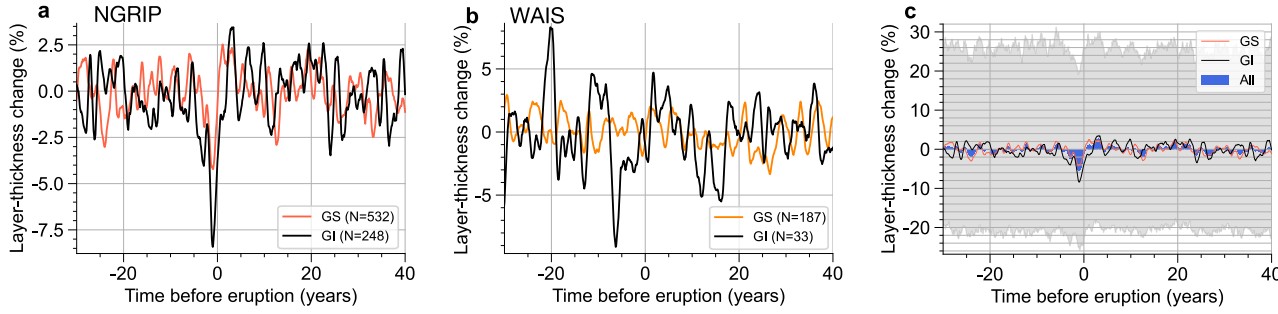

**Figure 10. a,b** Average layer thickness change after the volcanic eruptions of the unipolar data set during GI and GS in the NGRIP (**a**) and WAIS (**b**) core. The anomalies are defined with respect to the 50-year period before the individual eruptions. **c** Same as **a**, but in addition the average signal is shown in blue, and the gray bands show the 16- to 84-percentiles of detrended time slices covering individual eruptions.

polar response, but it also reflects the shorter return time of the eruptions in question compared to our unipolar data set (20 vs.

65 years). The corresponding maximum $\delta^{18}$O anomaly is 1.54$\pm$0.09 permil, derived from the youngest quarter of eruptions (Fig. 3c) to minimize the effect of diffusion. This yields $\frac{\gamma}{\alpha} = 0.037$ [0.030, 0.046] (confidence band via the above standard deviations). Assuming the present-day $\alpha = 0.8$, this would correspond to an accumulation sensitivity of 2.9 [2.3, 3.6] %·$K^{-1}$.

An alternative estimate is obtained by regression of the anomalies of individual eruptions. This is shown in Figure 11a, where a 4-year average accumulation anomaly (period of significant volcanic anomaly as determined from Fig. 10c) was used, along

with a 10-year average isotopic cooling anomaly (period of significant anomaly determined from Fig. 3c). The exponential CC relationship assuming $\alpha = 0.8$ is shown in green. Clearly, the large data scatter may permit a variety of functional relationships. Nevertheless, we perform a fit to Eq. 2, where we take into account noise in both variables. The noise levels can be estimated via the SNR, as explained in Sec. 3.1. We find SNR $= 0.24$ and SNR $= 0.44$ for $\Delta \log \lambda$ and $\Delta \delta^{18}$O, respectively. With the ratio of SNRs we can perform so-called Deming regression on the normalized data, which avoids underestimating the slope

as in regular linear regression (attenuation bias). This yields $\gamma/\alpha = 0.029 \pm 0.004$, and assuming $\alpha = 0.8$ the accumulation sensitivity is 2.4 [2.0, 2.7] %·$K^{-1}$, in agreement with a model-derived global sensitivity of 2.4 %·$K^{-1}$ (Bala et al., 2008). Note, however, that our given confidence interval does not reflect the significant freedom of choice in defining the average anomalies and performing the linear regression.

This accumulation sensitivity derived for the whole glacial ignores the clearly different accumulation reductions in GI and

GS, where GI eruptions lead to a more pronounced reduction (Fig. 10a,b). In contrast, if our $\delta^{18}$O analysis reflects a genuine state dependency of the regional temperature response, we expect the stronger Greenland cooling in GS to yield a larger accumulation reduction. In NGRIP, the peak accumulation reduction is 4.3$\pm$1.0% in GS and 8.5$\pm$1.5% in GI, while the peak $\delta^{18}$O anomaly is 1.8$\pm$0.1 and 0.78$\pm$0.23 permil, respectively (Fig. S17). If $\lambda$ were perfectly proportional to $\Delta T$ at constant $\gamma$, this would imply a GS-GI contrast of the isotopic sensitivity of $\frac{\alpha_{GS}}{\alpha_{GI}} = 4.5$ [2.2, 9.8]. This large difference may be unrealistic,

indicating that also the accumulation sensitivity may not be constant over time, as suggested by a previous analysis of the WAIS core (Fudge et al., 2016).

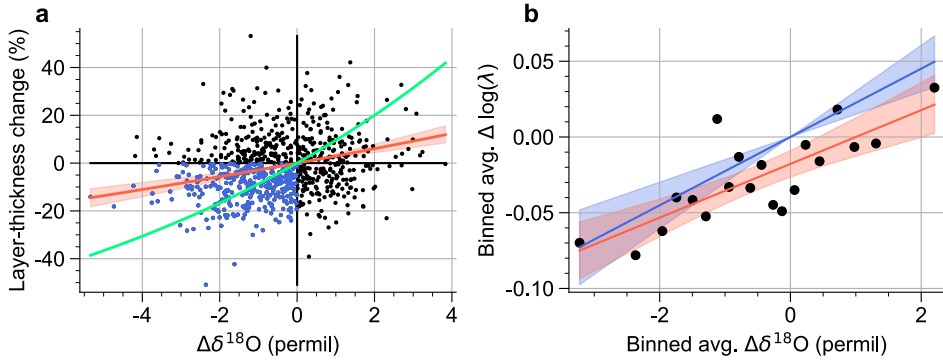

**Figure 11. a** Scatter plot of the layer thickness change and $\delta^{18}$O anomaly in NGRIP for the unipolar data set. The blue dots show the eruptions where both a negative $\delta^{18}$O anomaly and a layer thickness reduction is found. The red line and associated 95% confidence interval is given by exponentiating the linear Deming regression line of $\Delta \log \lambda$ and $\delta^{18}$O. A corresponding exponential CC relationship assuming $\alpha = 0.8$ is shown in green (see main text for more details). **b** $\Delta \log \lambda$ and $\delta^{18}$O anomalies averaged in bins, which are given by every 5th percentile of the $\delta^{18}$O data. A linear regression without (with) intercept is shown in blue (red).

In all above estimates of the sensitivity we assumed that a vanishing $\delta^{18}$O anomaly is accompanied by a vanishing $\lambda$ anomaly (as in Eq. 2). However, when reducing the noise level by averaging the data in $\delta^{18}$O bins we can see that the linear relationship does not pass through the origin (Fig. 11b). Thus, the response of either of the proxies includes one or more processes that are

not directly dependent on the underlying temperature change. This underlines that the true values and state dependencies of $\gamma$ and $\alpha$ cannot be revealed here. Nevertheless, on a qualitative level, the state dependency of $\Delta \lambda$ in GI versus GS, which is opposite to the state dependency of $\Delta \delta^{18}$O, strongly suggests a state-dependent climate response to volcanic eruptions, albeit of a more complicated nature than simple differences in the annual (regional or global) temperature response. The state-dependent accumulation rate reduction also makes it plausible that the seasonality of precipitation after volcanic eruptions may be altered

in a different way for GI and GS, which could in turn partly explain the differences in annual mean $\delta^{18}$O anomalies.

## 4   Discussion and Conclusions

Here we attempt for the first time to quantify the volcanic cooling following large eruptions during the last glacial period from ice-core data. This is done by precise alignment of two recent data sets of volcanism to high-resolution $\delta^{18}$O records from the same ice cores. Going back in time far beyond the observational and historical periods enables us to investigate the impact of

eruptions with very large return times. We find that the volcanic cooling signal is preserved in the ice-core records (Sec. 3.1 and 3.2), highlighting their potential to constrain the climatic impact of past volcanic eruptions in addition to tree ring and lake sediment records (Sigl et al., 2015; Tejedor et al., 2021). However, the preservation depends critically on a high measurement resolution of the $\delta^{18}$O records, a high accumulation rate at the ice-core site, and a moderate thinning of the annual layers (Sec. 3.3). Further, detecting a sharp multi-annual cooling relies on high resolution sulfur and conductivity records, which are

used to define the precise depths of the volcanic eruptions in the ice cores. Not all cores used here fulfill these criteria. Given

these limitations, we find that the observed isotopic anomaly after individual eruptions with centennial return periods is smaller than the high-frequency variability of the proxies (Fig. 1 and 2). The latter comprises multi-annual internal climate variability and non-climatic noise. As a result, we cannot give reliable estimates for the isotopic anomaly associated with individual eruptions and therefore also not estimate the cooling effect. Even the average anomaly at the time of deposition cannot be fully
reconstructed since the signal degrades over time in a way that is not well-understood (Fig. 3, Fig. 4a and Fig. S10).

With this caveat in mind, the amplitude of the Greenland isotopic response to bipolar eruptions during the younger half of the investigated time interval is consistent with their observed return period of 500 years, since the largest 4 eruptions of the last 2,000 years show roughly the same isotopic anomaly (Sec. 3.3). On the other hand, as the glacial records feature more diffusion, lower resolution, as well as less accurate alignment to the eruptions, the true isotopic anomaly should be larger.
Indeed, the youngest glacial eruptions in the larger unipolar data set show a clearly larger isotopic signal compared to the largest Common Era eruptions, despite a much lower return time of approximately 65 years (Fig. 3c). To better understand these differences, an in-depth comparison to eruptions of similar sulfate deposition covering the entire Holocene (Sigl et al., 2022) may be helpful in a future study.

Eruptions with larger sulfate deposition magnitude also lead to increased $\delta^{18}O$ cooling anomalies (Fig. 5 and Fig. 6a). Due
to the large noise levels, we cannot determine with confidence whether this relationship is linear. Future studies with larger data sets covering longer periods should be able to reveal whether eruptions with increasing return times simply have a linearly increasing amplitude and/or duration of volcanic cooling, or whether this relationship could be non-linear, and potentially have effects beyond a short-term cooling by compounding climatic regime shifts (tipping points). To do this it may be necessary to complement our methodology with idealized modeling of the proxy degradation over time.

By separating the data into eruptions occurring during the cold GS and milder GI periods, we find that the Greenland $\delta^{18}O$ anomaly is larger during GS, while on the other hand the anomaly in the Antarctic EDC core is larger in GI (Fig. 8). This suggests a state-dependent climate response with more pronounced Greenland (Antarctic) cooling following eruptions during GS (GI). If the state dependency is indeed robust, the pronounced Greenland cooling during GS eruptions may play a role in the apparent influence of bipolar eruptions on the transitions from GS to GI (Lohmann and Svensson, 2022). But our results
are also compatible with a more complicated difference in the climate response that is encoded in different sensitivities of the $\delta^{18}O$ proxy to the volcanic cooling. In addition, the volcanic forcing could be state-dependent, as a result of differences in atmospheric moisture and circulation, or of a modulation of the volcanic activity by the climate state (Cooper et al., 2018; Swindles et al., 2018; Farquharson and Amelung, 2022). We indeed find slightly larger sulfur deposition estimates in Greenland (Antarctica) during GS (GI) (Fig. 9). However, this cannot explain the state-dependent $\delta^{18}O$ anomalies, as shown in Sec. 3.6
by resampling the data such that the samples of eruptions during GI and GS have an equivalent distribution of sulfur deposition magnitudes.

It remains possible that the differences in $\delta^{18}O$ arise despite an identical climate response in GI and GS, for instance due to a fixed seasonality of the volcanic cooling (Robock, 2000) in combination with different seasonalities of precipitation for GS and GI (Steig et al., 1994; Werner et al., 2000; Li et al., 2005; Andersen et al., 2006). In particular, if there is less winter
precipitation in GS compared to GI, a less pronounced volcanic cooling (or even a warming) in winter compared to summer

(equally in GI and GS) would give a more depleted annual mean $\delta^{18}O$ signal in GS relative to GI. A similar situation could arise if there are different average precipitation source areas in GI and GS, for instance due to the differences in sea ice extent. If there is a latitudinal gradient in the volcanic cooling (Pausata et al., 2020), this could mean that the change in temperature gradient from source to sink after an eruption would be higher in GS, which also results in more depleted $\delta^{18}O$.

Due to the shortcoming of unknown glacial $\delta^{18}O$ sensitivity we also analyzed changes in accumulation rate after the eruptions (Sec. 3.7). While precipitation is generally believed to decrease proportionally to atmospheric cooling, we find that accumulation decreases in WAIS and NGRIP are clearly larger during GI eruptions, in contrast to the larger GS cooling suggested by Greenland $\delta^{18}O$. This reinforces that there is a kind of state dependency, but the opposing tendencies cast doubt on whether the larger GS $\delta^{18}O$ anomaly reflects more pronounced Greenland volcanic cooling. Since a vanishing volcanic 535 $\delta^{18}O$ anomaly does not coincide with a vanishing accumulation anomaly (Fig. 11b), it is clear that at least one of the two does not depend on temperature in a simple way. Just like $\delta^{18}O$, the local accumulation rate can be influenced by many factors apart from local temperature. Our analysis cannot reveal these factors, leaving the sensitivities of the $\delta^{18}O$ proxy and of the accumulation rate to temperature unknown. An extension of our analysis to other ice-core proxies may give further insights into the climate response. Besides the response, the actual climate forcing of large volcanic eruptions can be much more varied 540 compared to the simple surface cooling and drying assumed here, as evidenced by the recent Hunga-Tonga Hunga eruption (Millán et al., 2022).

Nevertheless, we provide a proof-of-concept to use ice-core proxy records in assessing the multi-annual climate response to volcanic eruptions, as well as its change with time and climate background state. The provided observational evidence of a state-dependent response of $\delta^{18}O$ and accumulation rate may be tested in studies with comprehensive climate models. Previous 545 modeling argues both for and against a state dependency of the global climate response to volcanic eruptions (Zanchettin et al., 2013; Berdahl and Robock, 2013; Muthers et al., 2015; Ellerhoff et al., 2022), as well as for state dependencies with opposite sign in a future warm climate (Fasullo et al., 2017; Hopfcroft et al., 2018). A state-dependency with respect to DO cycles has not been investigated yet. Studies considering volcanic eruptions in models that can simulate glacial DO-like switches between GI and GS states (Vettoretti and Peltier, 2016; Klockmann et al., 2020; Zhang et al., 2021; Kuniyoshi et al., 2022; Armstrong 550 et al., 2022) would be helpful. To test our results, direct comparisons with oxygen isotope-enabled climate models (Schmidt et al., 2007; LeGrande and Schmidt, 2009) should be performed, especially by extending previous studies on volcanic eruptions (Colose et al., 2016) to other climate background states. Understanding the spatial patterns of the precipitation and temperature response to volcanic perturbations at different latitudes (Colose et al., 2016), as well as the seasonality of the response (Zambri et al., 2017), may help explain our results. Further, comparisons of our results with data from non-ice-core archives are needed. 555 Tree ring records, for instance, are beginning to reach into the last glacial (Reinig et al., 2018; Pauly et al., 2020; Reinig et al., 2021) and may thus be used to assess the state dependency of the climate response to volcanic eruptions in future studies.

The presented methodology may also foster studies on climate variability and signal preservation in proxy records. Together with constraints on the strength of volcanic forcing, variability in climate records could be calibrated by the average volcanic climate response signal. Our preliminary analysis based on the signal-to-noise ratio suggests that the increase in the volcanic 560 Greenland $\delta^{18}O$ response during GS compared to GI is roughly the same as the increase in the non-volcanic proxy variability

(Fig. 4b). Assuming equal volcanic forcing, one might thus speculate that the much-discussed state dependency of climate variability inferred from Greenland ice cores (Ditlevsen et al., 1996; Rehfeld et al., 2018) is due to a state-dependent proxy sensitivity. But more detailed modeling of the proxy evolution over time is required to make a fair comparison between GI and GS states, as well as glacial and interglacial periods. Specifically, it would be insightful to model the post-depositional alteration and subsequent diffusion of an idealized volcanic cooling signal and compare this to the observed average signals reported here.

In summary, we show that multi-annual cooling after major volcanic eruptions is preserved in high-resolution $\delta^{18}$O records of polar ice cores. The inferred average $\delta^{18}$O anomaly remains smaller than the proxy variability, however. This may suggest that volcanism is not the main driver of multi-annual and decadal temperature variability during the last glacial, as opposed to what has been found from tree ring records during the Common Era (Sigl et al., 2015). However, the temperature change at the time of eruption is uncertain due to attenuation of the volcanic $\delta^{18}$O signal over time and an unknown sensitivity of the proxy. In addition, the glacial $\delta^{18}$O variability is inflated by non-climatic noise resulting from low accumulation rates. The Greenland $\delta^{18}$O cooling anomaly during the cold GS periods is larger than during the milder GI. The opposite holds for Antarctica. This may indicate that the climate response to the radiative cooling of the eruptions is state-dependent. But due to other effects, such as precipitation seasonality, it may also be the sensitivity of annual mean $\delta^{18}$O to the volcanic cooling that is state-dependent. Post-eruptive cooling is accompanied by a reduction in ice-core accumulation rates. In contrast to the pattern observed in $\delta^{18}$O, GI periods feature a larger volcanic accumulation reduction than GS. The mechanisms behind this complicated state dependency of the post-eruptive ice core signals cannot be revealed here. This may be achieved in future studies that test our observations in climate models, or analyse the volcanic signals in additional proxies. Further usage of the volcanic cooling signal to understand the climate variability implied by the $\delta^{18}$O proxy may also be fruitful, especially as larger volcanic data sets become available.

*Data availability.* The bipolar volcanic record is available in the supplementary material of Svensson et al. (2020), and the unipolar records are available in the supplementary material of Lin et al. (2022). The high-resolution oxygen ice core records of the individual cores are publicly available in the following online resources: NGRIP: http://iceandclimate.nbi.ku.dk/data/NGRIP_d18O_and_dust_5cm.xls; GISP2: http://depts.washington.edu/qil/datasets/gisp2_main.html; NEEM: https://doi.org/10.1594/PANGAEA.925552; EDML: https://doi.pangaea.de/10.1594/PANGAEA.754444; EDC: https://doi.org/10.1594/PANGAEA.683655; WAIS: https://doi.org/10.15784/601274. The GRIP record is available upon request from the corresponding author. The high-resolution sulfate records shown in the supplementary material are available in the following online resources: NGRIP: supplementary material of Lin et al. (2022); WAIS: https://doi.org/10.15784/601008; NEEM: supplementary material of Schüpbach et al. (2018); GISP2: https://doi.org/10.1594/PANGAEA.55537; EDC: https://doi.org/10.25921/kgv8-cn35. Code created for the statistical analyses can be found at https://github.com/johannes-lohmann/climpast_2023/.

*Author contributions.* J. L. designed and performed the research. J. Lin and A. S. analyzed the volcanic sulfur peaks. The paper was written by J. L. with input from all co-authors. All authors discussed and interpreted the results.

*Competing interests.* The authors declare no competing interests.

595   *Acknowledgements.* We thank V. Gkinis for help with the Antarctic high-resolution $\delta^{18}$O records. The project has received funding from the European Union's Horizon 2020 research and innovation programme under grant agreement No. 820970 (TiPES).

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
