# Peer review of "State-dependent impact of major volcanic eruptions observed in ice-core records of the last glacial period"

_EGUsphere, 2023_

## Author Response (AR1)

**RC1**

We thank the reviewer for the important comments to our manuscript. In our revised manuscript we addressed all the issues raised, and this significantly improved the content of the paper. The reviewer comments are given in bold, with our response below.

**- the paper presents new findings. while data is provided, code is not so the statistical robustness cannot be fully judged**

Code used for the statistical analysis is now available at: https://github.com/johannes-lohmann/climpast_2023

**- the literature review in the introduction should be expanded**

We included the references suggested below by the reviewer regarding modeling studies, and discuss these in the introduction of the revised paper.

**- possibilities to elucidate some of the challenges using non-ice-core evidence and/or climate modeling studies should be included**

We added a new segment in the outlook part of the Discussion Section describing the possibilities of using climate models (including isotope-enabled ones) as well as non-ice-core data (such as tree rings) to complement and explain our findings. Included here are citations to the references suggested by the reviewer.

**Detailed comments**

**- p1l18 here it would make sense to cite Pages2kConsortium2019 which demonstrates the clear multidecadal to centennial temperature variability induced by volcanic forcing over the common era in both models and using the Pages2k database**

Citation has been added here.

**-p2l30-35 This segment contains no references, and ignores that the changing nature of volcanic impacts in warmer-than-present climates -- and to some extent in colder climates such as the Glacial -- has received quite some attention in the literature. See e.g. Hopcroft, Kandlbauer et al., 2018; Ellerhoff, Kirschner et al., 2022; Aubry et al., 2021**

We agree that references were missing here, in order to complement those that were given previously in the Discussion section, where we noted there is both a presence and absence of modeled state dependency of the climate response after volcanic eruptions. We included the suggested references in a

new segment in the introduction, with special emphasis on the fact that modeling studies of a warmer-than-present climate have shown opposing state dependencies, i.e., both enhanced and reduced volcanic cooling. This helps motivate the need for more data-based studies.

**- Table 1: Please give an indication of the effective resolution of the WAIS data.**

We cannot give definitive numbers for this, but included relevant references and a new segment in the first paragraph of the Methods section 2.3 (then referred to in the Table caption), which summarizes the following.

The main effect to lower the resolution during measurement is mixing in the CFA system. This has been tested and documented in Jones et al (2017), and it appears that the effective "mixing length" is about 7mm, which is close to the measurement resolution provided in the data set (5mm). Thus, the effective time resolution of the raw measurements appears to be still almost as high as given by the measurement resolution in Table 1. But the true resolution of the isotope record as a temperature proxy is much lower due to diffusion of the isotopes. To be clear, here "resolution" refers to the degree of preservation of temporal isotopic variability as originally deposited on the ice sheet, and it is always lower than the measurement resolution.

The d18O diffusion length of WAIS is about an order of magnitude larger (Cuffey/Steig 1998; Jones et al 2017), and thus the true resolution, taking into account diffusion of the signal in the ice, is one order of magnitude lower (see also Jones et al 2018 for more on isotope diffusion in WAIS). But to give absolute values for this is difficult and beyond the scope of our paper.

In Table 1 we really only want to give an indication of the measurement resolution, because it is the first major potential limitation for a multi-annual cooling signal to be resolved, before diffusion in firn and ice. The main purpose of the table is to distinguish low-resolution (GISP2, EDC, EMDL) from high-resolution cores (all others), and to see how the measurement resolution decreases further back in time.

**- p6l170 Interestingly, volcanic cooling on the low end is consistent with the -.5 to -1K cooling for large eruptions found in Ellerhoff et al., 2022 Fig. 1a,b.**

This is indeed interesting, but it is hard to compare the two studies quantitatively. We don't know the isotope sensitivity after all, and we cannot match the magnitude of the eruptions in question with the low-end cooling events in Ellerhoff et al. They don't give concrete, absolute numbers of the volcanic forcing and associated volcanic cooling, which we could compare our isotopic cooling to (assuming some isotopic sensitivity). Finally, they include both volcanic and time-varying solar irradiance forcing at the same time.

**- p8l1 The wording 'signal-to-noise-ratio' is strange to identify a peak over internal variability. Is it the same as the Mean Standardized Anomaly e.g. used in the abovementioned study? Perhaps that would be a better name.**

Our signal-to-noise-ratio is very similar, but not exactly the same as the mean standardized anomaly. In Ellerhoff et al, a 1-year average volcanic signal is normalized by the standard deviation of the entire time series (including other volcanic periods) at a given grid point. We are only comparing averaged time series segments with and without volcanic eruption of the same short length (e.g. 6-year averages). We also only use "local" means, subtracting the mean of the previous 50 years and not the entire record.

Regarding the terminology: We did mention in the manuscript that it is not a typical signal-to-noise-ratio of the temperature proxy in absolute terms, but we were not clear how the term is to be understood. In the revised manuscript we added an explanation thereof in Section 3.1.

The term "mean standardized anomaly" may be problematic in our study, because we actually don't know what the anomaly is standardized/normalized by. We do not have a modelled or observed temperature signal, but a signal that comprises natural climate variability, precipitation intermittency, post-depositional/stratigraphic noise, and more, in unknown proportions. The sharpness and amplitude of the volcanic peak over the background - and thus the signal-to-noise-ratio - is furthermore determined by synchronization uncertainties, the width of sulfate peaks, and the averaging introduced by the measurement resolution.

For these reasons, our signal-to-noise-ratio is not exactly a "peak over internal variability". For any signal-to-noise-ratio one has to define a priori what should be considered noise and what should be considered signal. Here we aim to extract the volcanic cooling signal only. In this context also climatic, internal variability is unwanted noise, as it introduces uncertainty in the signal we try to extract.

**- Outlook: Modelling studies addressing climatic and environmental effects of volcanic eruptions should provide interesting background that is neglected here so far. In particular including isotope-enabled GCMs could be informative, see e.g. Ellerhoff et al. 2022 for a conceptual studies for the Glacial, or the work of Allegra LeGrande and coworkers.**

We added references from the suggested authors and discuss in Section 4 the prospects of isotope-enabled models and previous study investigating spatio-temporal response patterns of temperature and precipitation in order to explain the volcanic ice core signal.

**- p13 section 3.5 The discussion of state dependency could be a bit better introduced, and generally the wording could be more careful. For example, the difference between local (e.g., Greenland) and global/hemispheric climate is blurred, and the common definition of climate sensitivity is broadened without conceptual introduction. Indeed, stronger isotopic cooling in the Glacial vs. the common era could indicate local climate sensitivity to volcanic forcing changed, or the proxy sensitivity changed, or global climate sensitivity changed. Ellerhoff and coauthors suggest there is no strong state dependency in the response to volcanic forcing between the Glacial and Preindustrial.**

We agree that the differences of regional and global climate sensitivity, as well as proxy sensitivity were not clearly outlined. In the beginning of Section 3.5 of the revised manuscript we now distinguish "climate sensitivity" as the short-term temperature response to the volcanic radiative forcing from what is commonly meant by climate sensitivity (global T response to instantaneous, permanent doubling of $CO_2$ and subsequent equilibration). We further added a distinction of state dependency of global climate sensitivity from regional climate sensitivity, i.e., changes in the spatial pattern of the temperature response. Finally, we contrast this with state dependency of the proxy sensitivity, which is not dependent on any differences in the temperature response. Later in the revised text, we specifically refer to regional and global climate sensitivities where necessary.

Indeed the study by Ellerhoff et al suggests no state dependency, which we now mention explicitly in Introduction of the revised manuscript. Nevertheless, it is a modeling study analyzing temperature, whereas we analyze isotope data, and thus it remains to be shown what causes the discrepancy. As mentioned more explicitly in the Introduction and Discussion section of the revised paper, other modeling studies do find state dependency. We cannot speculate the specific reasons for this, but different models may have different biases in feedbacks, or may be missing physics that are responsible for a potential state dependency in the real world.

**-p3.6 Schindlbeck-Belo and coauthors do not find a state-dependency of volcanic forcing in their reconstruction for the last Glacial Cycle (ESSDD, 2023), rather only an amplification during the deglaciation.**

This is interesting, and if the paper is published in time we will include a citation. On the other hand, from the abstract it seems that the methodology and data sets may not be ideal to address the question of state-dependency. Quote from the abstract: "*To correct for the incompleteness of the tephra record we include stochastically generated synthetic eruptions, assuming a constant background eruption frequency from the ice core Holocene record.*"
So it seems like to some degree a state-independency is built in.

**References**

Cuffey, K. M. and Steig, E. J.: J. Glaciol., 44, 273–284, 1998

Jones, T. R., White, J. W. C., Steig, E. J., Vaugh, B. H., Morris, V., Gkinis, V., Markle, B. R., and Schoenemann, S. W.: Atmos. Meas. Tech., 10, 617–632, 2017

Jones, T. R., Roberts, W. H. G., Steig, E. J., Cuffey, K. M., Markle, B. R., and White, J. W. C.: Nature, 554, 351–355, 2018

**RC2**

We thank the reviewer for thorough evaluation of our manuscript. We addressed all of the reviewers comments, which led to a significant revision of the text, and clearly improved the paper. Below is our response to all reviewer comments (given in bold).

**The paper is well written even though it would be easier to follow with a simpler and straightforward writing style.**

We gave the text an overhaul and tried to simplify the writing style where possible.

**It also refers very often to SI that are central to the study, so that the reader has to stop and go back and fourth through various documents at almost every paragraph to follow the results and reasoning presented in the main manuscript. I however acknowledge that a lot of work has been developed in an attempt to use appropriate various statistical methods to analyse a large set of ice core records.**

We agree that too much material was relegated to the SI, and we moved the most important supplemental figures to the main text (S9, S10b, S13, S17). The remaining supplemental figures have a more supporting role, and the conclusions drawn from the figures are sufficiently summarized in the main text with one sentence. These figures either demonstrate the absence of a certain effect (S4, S5, S6, S7, S10a), such as a very weak correlation, or they confirm what is shown in the main text for other ice cores (S3, S8, S12, S16), or they give details related to the methodology (S1, S2, S11, S14, S15).

1. **If I understood correctly, the authors work on estimated bipolar and unipolar sulfate depositions magnitude averaged across all cores where individual eruptions have been identified. I understand that this approach is meant to improve the signal to noise ratio but I wonder if it is not actually the opposite that is obtained with such procedure.**

Indeed, our work is based on the unipolar and bipolar volcanic datasets that were published by Lin et al., CP, 2022. In this publication, the sulfate deposition in each ice core is listed for each eruption, and the relative sulfate deposition in each core is discussed. Greenland ice cores are synchronized using the shared volcanic acidity signal (eg. Seierstad et al., QSR, 2014) and the same is true for Antarctic ice cores, so we are certain that the averaged sulfate depositions are originating from the same eruption. For the unipolar dataset we can compare the relative sulfate deposition within Greenland or within Antarctica, but we cannot know the magnitude of the eruption as it depends on its location relative to the ice sheet. The list of bipolar volcanic eruptions was originally published in Svensson et al., CP, 2020. For the bipolar eruptions we are certain that the sulfate from the eruptions made it to the stratosphere. In order to compare the magnitude of the bipolar eruptions we need to estimate the stratospheric sulfate injection that is calculated from both the Greenland and the Antarctic sulfate

depositions (Lin et al., CP, 2022). It is thus only the bipolar eruptions for which the estimated magnitude can be compared to those of well-known eruptions, such as Tambora or Salamas.

It is a common approach to average the sulfate deposition for several cores within Greenland or within Antarctica in order to improve the estimate of the eruption magnitude (e.g. Sigl et al Nature 2015). A priori it seems very reasonable to assume that averaging the sulfate deposition estimates across all cores of one Hemisphere would improve the estimate of the deposition magnitude. From Lin et al 2022 (and older studies) it becomes clear that the deposition in an individual core is quite unreliable. This is because the quantitative differences in sulfate deposition of the same eruption in multiple cores are very large, and even more because eruptions with very large sulfate deposition in one core are commonly missing in another. This has been demonstrated explicitly in a study by Gautier et al (2016) for low-accumulation sites, where replicate cores were drilled only 1m apart. There it was found that there is a 30% chance of an eruption missing, and there is an uncertainty of 65% on the volcanic sulfur flux.

Even though this issue relates more to Lin et al 2022 than to our study, we have dedicated two new paragraphs in Section 2.1 to discuss this, and further give two new SI figure that illustrate the high variability of the deposition estimate for the same eruptions in different cores.

The study of Svensson et al 2020 shows that even bipolar eruptions are often missing in multiple cores. In the caption to Fig. 1c we give this information: Out of the 82 eruptions, there were 49 eruptions with four cores, 14 (10) with three (two) cores, and 9 represented by NGRIP only. Large eruptions missing in one or more cores could be due to:

a)
A highly variable volcanic ice core sulfate concentration, which often falls below the background level. This in turn can be due to an acually spatially heterogeneous deposition, or also due to snow redistribution and uneven layer thinning.
b)
Deficiencies in the sulfate measurements and resolution for some cores, and limitations to the synchronization across the cores of one Hemisphere that makes it difficult to identify some eruptions in some cores and during some periods of the record.

Given the deposition is so variable for even relatively close-by cores motivates averaging the deposition values. In terms of the climate impact one can a priori assume that a more large-scale metric (and not the local deposition at the exact ice core site) would be appropriate. Averaging is even commonly done across Hemispheres (e.g. Sigl et al Nature 2015), and used to construct global aerosol forcing records for driving Common Era CMIP climate simulations.

Note that we do actually show some results where we compare the usage of deposition data from individual cores with the usage of all cores (Fig. 4a and Fig. S12). This was done because when using the deposition of one core instead of the averaged deposition we found a better correlation with the isotopic anomaly in the same core. From the paper on page 11:

*"There is generally a better correlation of the anomaly with the deposition in the individual core, and not the deposition averaged over multiple cores (Fig. S12)."*

In the manuscript we speculate this is because of more precise depth estimates for eruptions that actually show up as large sulfate spikes in a given core, and thus give a very good alignment of the isotopic record and the true eruption depth. So the reviewer may be right that there could be situations where using the averaged deposition is not as good as using the deposition value from one core. But it remains slightly puzzling why this can be the case, and from the reviewer comment we cannot deduce any other suggested reasoning besides the one already given in the paper. For the GS/GI resampling (Fig. 5a,b,e,f) we chose the average deposition data, but there is no qualitative differences when using the deposition from the individual core.

Another potential reason against averaging one could bring up is that the mean deposition is different across the ice cores. Thus, if one core with, for instance, much larger average deposition is only present for some of the eruptions, it will introduce some variations in the mean signal, leading to higher values whenever the eruption has been recorded in the core. This introduces statistical noise into the averages, which is now mentioned in the revised paper. It is certainly the case to some degree, but from our new SI figures it can be deduced that the differences in the mean deposition are much smaller than the variability of the deposition values for the same eruptions across cores. This means that it is still better to average, despite differences in the mean, and in the revised paper (Section 2.1) this is the reasoning we give for why we in general prefer the averaged deposition.

**Little information is given on the form of the full distribution of the sulfate depositions magnitude from each cores for individual eruption. This analysis would help understand the magnitude distribution of sulfate deposition for each eruption across used ice cores (Northern and/or Southern). Such information might turnout to be important to address the objectives of the study by helping stratify (relying on the full distribution) small (smaller that Pinatubo) from larger eruptions of different scale (Tambora, Samalas or even Toba-like eruptions). It is indeed difficult for me to fully grasp the significance and physical relevance when it comes to climate sensitivity or temperature anomalies deduced from the oxygen isotopic signal when working on such large compound of selected events that are simply averaged.**

We cannot determine from this comment what exact distributions the reviewer would like to see. But in the revised paper we included more analysis on the quantitative discrepancy of deposition from individual eruptions across cores, which also relates to the previous comment. One new SI figure gives correlation coefficients and scatter plots for the sulfate deposition in pairs of cores. Further, we added a SI figure with distributions as in Fig. 6, but for the individual cores. This should allow the reader to judge the variability in sulfate deposition across cores, and help motivate the averaging that was perfomed. For more, the reader is referred to Lin et al CP 2022, as a detailed analysis of the spatial distribution of deposition, including a statistical inference of its distributional form, is beyond the scope of this paper.

Note that we cannot distinguish smaller-than Pinatubo eruptions, since in Lin et al 2022 a sulfate deposition threshold has been applied, which in Greenland is about half the volcanic sulfate deposition of Tambora in 1815 CE and in Antarctica is comparable to the Pinatubo 1991 CE eruption.

Our "compounding" of events is quite standard. We simply compute the response of an eruption of a certain cumulative magnitude class - that is, all eruptions of magnitude higher than a certain threshold - by averaging over the response of all individual eruptions of the class. So the significance of the signal deduced from the averaged oxygen isotope signals is clear: Assuming the data sets are close-to-complete (not missing too many eruptions), we can for instance report the average isotopic anomaly for a 1-in-500 year eruption (this corresponds to the return time of the bipolar data set). As discussed in the manuscript, the cooling signal in absolute terms remains unknown, due to the signal degradation over time, and because of the unknown proxy calibration. We do acknowledge that every eruption is born different, and that it could lead to a variety of climatic impacts. This is mentioned in the Discussion with the example of the Hunga-Tonga Hunga eruption. So the reviewer is right that the volcanic data sets used here are compounds of a large number of eruptions of a potentially quite variable nature. In fact, if they are close to complete, the data sets cover essentially any eruption that led to an above-threshold sulfate deposition, occurring in the period 11.7-60ka. It would be ideal to stratify according to further criteria, but besides the sulfate deposition we have at present no information on the eruptions that would allow us to do so.

Indeed, the unipolar sets are quite heterogeneous, as they include smaller eruptions which nevertheless led to a large sulfate deposition due to close proximity to the ice sheet. But the sheer size of the data sets, and the associated reduction of the noise in the average signal, still warrants us to construct an average signal, and it allows us to separate the data into subsets that are more finely stratified according to age, deposition magnitude, and climate background state. As mentioned above, the true magnitude (as compared to known historic eruptions) should only be considered for the bipolar eruptions. So in the unipolar case we cannot speak of 1-in-X year eruptions in the global sense, but only of 1-in-X year eruptions in terms of the local sulfate deposition at the ice core site, which also includes deposition from non-stratospheric eruptions. In this case it is true that the physical significance of the signal in absolute terms is unclear. But we can still make relative comparisons of different subsets of eruptions, either comparing intervals in time, climate states, or deposition magnitude classes. These relative comparisons are the main focus of the study. For instance, a large part of the study focuses on GI vs GS, which is done in an age-controlled and magnitude-matched sample. For this, the unipolar data sets are very well suited.

Regarding the stratification of smaller versus larger eruption, we added additional analysis which very clearly shows the influence of eruption magnitude in the bipolar data set. We previously had a scatterplot and correlation in Fig. S10 for the Greenland stack, which showed that there is indeed a weak correlation, with larger magnitude eruptions producing a larger isotopic cooling on average. In addition to this, in the revised paper we added a new figure that shows the bipolar response curves in

the Greenland stack and EDC, coarsely stratified in magnitude according to the aerosol loading. The results are discussed in a new paragraph at the beginning of Section 3.4.

2. **On the same line, we can see on the Fig. 1c for example, that the pdf distribution is hardly significantly different from the back noise, with as much positive as negative isotopic excursions. This points out that the selected events are not all corresponding to an actual volcanic eruption and also that working with only one moment of the distribution might not be the best way to analyse these datasets.**

We understand this impression of the reviewer, since the blue distribution in Fig. 1c still generously spills over to positive values. But it is not true that the distribution is hardly significantly different from the background noise (gray distribution), and that there should be as many positive as negative values. The individual values are given by the short vertical lines, and it can be seen clearly that there are many more negative excursions. To be precise, 23 out of the 82 bipolar eruptions feature a positive anomaly. This number is now mentioned in the revised manuscript. Any statistical test would show that the difference in distributions is highly significant. In colloquial terms, given the gray distribution is essentially symmetric, it is highly unlikely to draw a random sample of N=82 from the gray distribution and find only 23 values to be positive, let alone find so many eruptions in the far tail (and beyond) of negative values.

Unfortunately, an interpretation and comparison of the second and higher moments of these distributions would not be without problems, since the synchronization of the records comprising the stack is naturally better for the volcanic slices compared to the non-volcanic slices. So for the non-volcanic slices some true climate variability may be averaged out when stacking the non-perfectly aligned records. This could be partly responsible for the gray distribution to appear more narrow.

We hope to change the reviewer's interpretation - i.e., the idea that because the distribution does include quite a lot of positive values should mean many of the events are not actually volcanic eruptions - by the following clarifications, and according revisions to the text (mostly Sec. 3.1):

a)
As can be easily seen from the gray distribution in Fig.1c, there is a large non-volcanic variability of the proxy, which comprises both proxy noise and internal multi-annual temperature variability leading to fluctuations in the mean of up to 1 permil. With previously established (though uncertain) calibration of the isotopic proxy this easily corresponds to 1K or more of variability (in the revised text we derive the numbers 1.25-1.45 K). It is thus clear that a volcanic cooling of 1.0 K (corresponding to a very large eruption) can very easily be "swallowed" by a random, positive excursion of the non-volcanic proxy variability, leading to a positive multi-annual isotope excursion despite the volcanic cooling. It is furthermore not a priori given that every volcanic eruption should lead to a cooling in Greenland and Antarctica. Suggesting a positive d18O anomaly means that there was no volcanic eruption is thus not a valid argument, unless additional information would be available.

b)
The volcanic eruptions were chosen in previous studies based on sharp sulfate spikes. This is a well-established procedure, and it does not require a novel validation with potentially negative d18O anomalies. Our study is indeed building on the assumption that large sulphur spikes in ice cores are of volcanic origin. It is an assumption that seems justified by the volcanic eruption history of the last 2000 years. Unfortunately, we are not able to work on individual eruptions in the last glacial period as the isotopic noise is generally larger than the cooling signal, as pointed out by the reviewer. That is why we work on averaged assemblies of eruptions only.

3. **It would also be important to explain which criteria have been used to select the "identified individual eruptions". How are authors sure that the sulfate deposition that is selected corresponds to a stratospheric large eruption?**

This is the main topic of the Lin et al., CP, 2022 publication. For the unipolar datasets we cannot be certain that the identified eruptions are stratospheric, as some of them will be 'local tropospheric' eruptions, but for the bipolar eruptions we are quite convinced that the eruptions have been stratospheric, otherwise the sulfate deposition would not have occurred in both hesispheres.

4. **Coming to the climate sensitivity under GS and GI the climate backgrounds, again I wonder how much the conclusions drown are not dependent on the eruption selections and averaging procedure questioned in my previous comments. The authors do not clearly give the number of events selected in the GI and GS or show the full distribution of sulfate deposition from individual ice cores for each identified event.**

The conclusions do not depend on these issues. We do not "select" eruptions in GI and GS, but we use all eruptions available from the continuous records and simply separate them depending on whether they occur during GI or GS. The original volcanic data sets (Svensson et al 2020, Lin et al 2022) employ the same method for identifying eruptions during GI or GS, and thus we don't see much potential for non-robust conclusions, except those caveats already mentioned in the manuscript, which cannot be addressed using currently available data. Due to the high noise levels averaging is strictly necessary to draw any conclusions, and it is done rigorously in fact. We sample eruptions occurring during GS and GI such that they obey the same distribution of sulfur deposition magnitude, and which correspond to the same time period within the glacial. This is what is necessary to get a robust result.

That being said, as discussed in detail in the manuscript, we cannot be sure that two eruptions occurring in GI and GS, respectively, and which are equal in their ice core sulfate concentration, actually correspond to eruptions with the same total amount of stratospheric aerosols. The amount of deposition is modulated by the hydrological cycle and the atmospheric circulation pattern, which could be different in GI versus GS. But these factors cannot be controlled for in the present study.

The reviewer correctly points out that we ommitted the number of GI and GS events in some cases. This is added to the revised manuscript. Regarding the distribution of sulfate deposition, as mentioned above, we included further analyses on the deposition magnitude distributions and the differences in deposition across cores.

**All in all, I find the work achieved in the present paper very useful and timely but it would be important to assess that obtained conclusion are not dependant on the selected events and method used to build the compounded events.**

We cannot assess this. Our methodology is only possible with a volcanic data set derived from ice cores, and the data sets used here are the only continuous data sets of volcanism during the last glacial period. It is a misconception that our study uses "selected events". The present ice core volcanism reconstructions are the best available efforts constructing a continuous record of all eruptions during the period in question that are above a certain magnitude threshold, either in terms of the unipolar sulfate deposition (Lin et al 2022), or in terms of the bipolar deposition (Svensson et al 2020).

Admittedly, the data sets are still imperfect, and, within the methodology of the two studies mentioned, there would be different ways to construct the data sets, for example by changing the threshold on the sulfur background level. But it is far beyond the scope of our paper to construct alternative volcanic data sets in order to test whether they will lead to different conclusions regarding the isotopic response. This would yield entirely new data sets that need to be validated and analyzed.

Ongoing work will yield extended data sets covering the whole last glacial, and the entire Holocene, and an analysis of these data sets is planned in order to confirm the robustness of our results.

**It is hard to understand any significant isotopic signal or temperature anomalies or even climate sensitivity with a mean signal.**

We agree that more information besides the mean signal could in principle be extracted from the data. But given the large noise levels and non-volcanic variability in the proxies, using the mean signal over a large number of eruption is the most relevant and meaningful method at this stage of the research.

Besides what is already given in the manuscript (investigating the influence of unipolar deposition magnitude, relative bipolar deposition, time period, as well as climate background state), we do not have any further information on the nature of the eruptions or the climate conditions at the time of eruption that would allow us to separate them and to use quantities besides the compound mean signal. As mentioned in a previous comment above, the main focus of the study is not to estimate temperature anomalies and climate sensitivities in absolute terms - since this is not possible due to unknown proxy calibration and signal degradation - but to assess the relative strength of the signal for different magnitude classes, time periods, and climate background states. This is done by comparing the mean signal in subset of eruptions, while carefully accounting for changes in other factors, as done in the paper.

**This is even more important since previous work have shown that the temperature response is not linearly related to the magnitude of the eruption due to self-limiting microphysical processes in the stratosphere and internal climate feedbacks.**

This is a very interesting point indeed. The paper already presents an assessment of the potential linearity of the response to the magnitude, but, as mentioned in the manuscript, we cannot rule out a non-linear relationship since the spread in the data is too high. This is certainly an exciting avenue we want to pursue in further research as larger and more detailed data sets become available.

**I would suggest stratifying as much as possible, the eruption according to there magnitude before doing all the analyses presented in this study. I don't see any other way to address and investigate the climate responses to eruptions occurring during the last glacial period.**

We hope our responses to all comments above can convince the reviewer that our methodology and results, together with the new additions to the manuscript, provide a very reasonable basis to assess and investigate the climate response, given the uncertainties and limitations discussed. The paper presents an analysis of the data stratified into magnitude classes for most relevant cases (given the current sampe size), and a new analysis of large-magnitude versus smaller-magnitude bipolar eruptions has been added to strengthen the existing analysis.

**References**

Gautier, E., Savarino, J., Erbland, J., Lanciki, A., and Possenti, P.: Clim. Past, 12, 103–113, https://doi.org/10.5194/cp-12-103-2016, 2016.

Lin, J., Svensson, A., Hvidberg, C. S., Lohmann, J., Kristiansen, S., Dahl-Jensen, D., Steffensen, J. P., Rasmussen, S. O., Cook, E., Kjær, H. A., Vinther, B. M., Fischer, H., Stocker, T., Sigl, M., Bigler, M., Severi, M., Traversi, R., and Mulvaney, R.:, Clim. Past., 18, 595, 485–506, 2022

Seierstad, I. K. et al., Quat. Sc. Rev., 106, 29–46, 2014

Sigl, M. et al., Nature, 523, 543–549, 2015

Svensson, A. et al., Clim. Past, 16, 1565–1580, 2020